# SELFISH EMERGENT COMMUNICATION

## ABSTRACT

Current literature in machine learning holds that unaligned, self-interested agents do not learn to use an emergent communication channel. We introduce a new sender-receiver game to study emergent communication for this spectrum of partially-competitive scenarios and put special care into evaluation. We find that communication can indeed emerge in partially-competitive scenarios, and we discover three things that are tied to improving it. First, that selfish communication is proportional to cooperation, and it naturally occurs for situations that are more cooperative than competitive. Second, that stability and performance are improved by using LOLA (Foerster et al., 2018a), especially in more competitive scenarios. And third, that discrete protocols lend themselves better to learning cooperative communication than continuous ones.

## 1 INTRODUCTION

The principles involved in the evolution of effective communication are essential for artificial intelligence research since they may lead to innovative communication methods for use by interacting AI agents and multi-robot systems. AI agents need a common language to coordinate with one another and to communicate successfully with humans (Wagner et al., 2003). The emergence of communication protocols between learning agents has seen a surge of interest in recent years, but most work tends to study fully-cooperative agents that share a reward (Foerster et al., 2016; Havrylov & Titov, 2017; Lazaridou et al., 2016). Work on selfish agents, that separately optimise their own reward, has been limited, with results suggesting that selfish agents do not learn to use a communication channel effectively (Cao et al., 2018; Jaques et al., 2018). This has contributed to a perspective in the field that emergent communication is a purely cooperative pursuit Lanctot et al. (2017). This is in contrast to theoretical results in game theory that show it is possible to use an existing cheap-talk communication channel effectively to resolve situations of partial conflict (Farrell & Rabin, 1996). We aim to reconcile these different findings and establish the degree of cooperation necessary for useful communication to emerge.

To study this problem in detail, we look at the simplest case of communication: a sender-receiver game (Lewis, 1969). This is a game of incomplete information, wherein a sender obtains private knowledge and communicates this to a receiver via a signal, or message. The receiver uses the message to inform its action in the environment. Though messages are arbitrary, and initially meaningless, the players can coordinate upon a conventional meaning for the signal (Skyrms, 2010). While the classic game is fully cooperative, we introduce an arbitrary level of conflict between our two players and investigate whether communication can emerge for each level of conflict. Contrary to current literature in machine learning, we find evidence that communication emerges in competitive scenarios—provided that the agents' interests are at least partially aligned. We further find that using LOLA (Foerster et al., 2018a)—an effective strategy for resolving social dilemmas—to explicitly model opponent learning yields more effective communication protocols. Finally, we consider the difference between continuous and discrete emergent communication and find that discrete communication lends itself better to cooperative communication.

## 2 RELATED WORK

### 2.1 FULLY COMPETITIVE/COOPERATIVE MARL

Multi-agent RL is used for many fully-competitive, zero-sum games (Silver et al., 2017; Vinyals et al., 2019; OpenAI, 2018) with the objective of finding a single, best *learned* agent (or team) at test time. In these games, a "best" agent is one that can outplay all opponents (Balduzzi et al., 2018). We cannot evaluate its play by looking at the reward that agent achieves against a given opponent because the reward received will be a function of the choice of opponent.[1] In contrast, MARL in fully cooperative games has usually been able to make the assumption that we get to pick our team, and so can use self-play to try and achieve a maximal possible reward. (Foerster et al., 2018b). In this way, fully-cooperative MARL compares the maximum performance between *learning algorithms and architectures* by optimising the joint reward of the team.

### 2.2 GENERAL-SUM MARL

We investigate the space of partially competitive games known as general-sum games, where there is some amount of common interest and some amount of conflict. In this case, care must be taken in defining the "best" agent: it is not necessarily the agent that does as well or better than all opponents because, to achieve the highest possible reward in a general-sum game, agents might have to cooperate to some extent (Leibo et al., 2017). For example, consider an agent playing iterated prisoner's dilemma. The agent that always defects will never have a reward worse than its opponent, but, when playing against a tit-for-tat agent, it will also not achieve the total reward of an agent that is also playing a tit-for-tat strategy.

Since the maximum expected reward may only be possible by cooperating, agents must learn how to coordinate with each other. This can be done by training together or learning to understand opponent intentions at test time by observing their actions. The latter allows for ad-hoc comparison of *learned agents* at test time; the former requires comparing *learning algorithms* trained together. The latter should then require a sequential/iterative game, so there is time to infer opponent intentions before acting (Fujimoto & Kaneko, 2019). It may also require meta-learning or other modifications to understand and adapt to these intentions as it seems current self-play methods are insufficient to adapt to ad-hoc play even against different versions of their own architecture (Bard et al., 2019). Work in general-sum MARL has mostly worked on analysing *learning algorithms'* ability to cooperate and resolve social dilemmas (Foerster et al., 2018a; Letcher et al., 2019; Lerer & Peysakhovich, 2017)

### 2.3 EMERGENT COMMUNICATION

To date, investigations of emergent communication have remained mostly in the realm of fully-cooperative games (Lazaridou et al., 2018; Das et al., 2019; Evtimova et al., 2017). For continuous communication, Singh et al. (2018) claimed to learn in mixed cooperative-competitive scenarios. However, their setup uses parameter sharing between opponents; their "mixed" case is non-competitive (and implicitly cooperative); their competitive game is actually two stages—one fully cooperative and one fully competitive; and, their results in the competitive scenario are simply to mask out all communication. Previous attempts to learn discrete emergent protocols by selfish agents in competitive games have failed (Cao et al., 2018) unless additional, more complex learning rules are adopted (Jaques et al., 2018). In the latter case, they compare *learning algorithms* trained together as opposed to *learned agents* at test time to avoid a significant issue of comparing emergent communication agents—i.e., different protocols. Since an emergent protocol only has meaning between the agents that learned it together, comparing two learned agents at test time would require that they infer each others' protocols without training with them. This seems impossible, and it is more reasonable to frame it as a meta-learning problem where agents can have a brief adaptation period to synchronise their protocols. Meta-learning is beyond the scope of this paper, so we follow Jaques et al. (2018) in comparing *learning algorithms* trained together. We also take a more principled approach than previous work by guaranteeing no communication through the action space,

---

[1]For multi-agent RL, we must define the space of possible agents that we play against because our aim is not merely to beat one opponent but all potential opponents—e.g., in non-transitive zero-sum games (Balduzzi et al., 2019). Following the "AI agenda" of Shoham et al. (2003), we believe it is reasonable to consider a space of learning agents similar to our own that are trained with gradient descent.

using more rigorous, quantitative criteria for evaluating communication, and precisely setting the levels of competitiveness. Notably, this work does not propose new architectures or learning rules but aims to take a critical look at existing beliefs and draw important distinctions as Kottur et al. (2017) did for natural language emergence.

## 2.4 SENDER-RECEIVER GAMES

Our experiments use the simple emergent-communication framework known as a sender-receiver game (or "referential game" (Lazaridou et al., 2016)), which finds extensive use in economics (Riley, 2001) and philosophy (Lewis, 1969; Skyrms, 2010) among others. In the classic game, the sender is given a target value to be communicated to the receiver via a message. The receiver receives the sender's message and must decode it to predict the target value. Both players are rewarded according to the negative of the receiver's prediction error. In this fully-cooperative setting, players often coordinate a protocol to transfer information as effectively as possible (Skyrms, 2010). The messages between the sender and the receiver can be categorised as "cheap talk": messages are costless, non-binding, non-verifiable and *may* affect the receiver's beliefs (Farrell & Rabin, 1996).

Our work benefits greatly from Crawford & Sobel (1982) – a seminal work in classical game theory. They study possible *fixed* communication equilibria under competition by giving the sender and receiver different targets and creating a conflict of interest. They perform a static analysis and prove the existence of a Nash equilibrium where the amount of information communicated is proportional to the alignment between the players' interests; however no informative equilibrium exists when interest diverge too greatly. In contrast, we do not look for existence of an equilibrium but do a dynamic analysis and show the *feasibility* of communication using standard learning rules in RL. We do not explicitly aim for equilibria (Shoham et al., 2003) but look at the information transfer of communication protocols in flux (and therefore out of equilibrium). This is more in line with previous work in emergent communication (Jaques et al., 2018) as well as evolutionary signalling (Skyrms, 2010).

## 3 THE CIRCULAR, BIASED SENDER-RECEIVER GAME

To investigate a range of competitive scenarios, we introduce a modified sender-receiver game with a continuous-bias variable, $b$, that represents the agents' conflict of interest, ranging from fully cooperative to fully competitive. The two players—the Sender ($S$) and the Receiver ($R$)—have corresponding targets ($T_s$ and $T_r$), which are represented by angles on a circle that are $b$ degrees apart: $T_r = (T_s + b) \mod 360°$.

The game starts with the sender's target being sampled uniformly from the circle $T_s \sim$ Uniform$[0, 360)$. The sender is given its target as input and outputs a message, $m = S(T_s)$, consisting of a single, discrete token from a vocabulary $m \in V$. The receiver is given the message and outputs a scalar action $a = R(m)$. The goal of each agent is to make the receiver's action as close as possible to its *own* target value. After the receiver acts, both players get a loss between the action and their respective targets, $L^i = L(a, T_i)$. By using an $L_1$ loss between the angle of the target and action $L_1^i(T_i, a) = \min(|T_i - a|, 360° - |T_i - a|)$, it is evident that a game with $b = 0°$ is fully cooperative ($L_1^r = L_1^s$) and a game with the maximum bias $b = 180°$ is fully competitive or constant-sum (a generalisation of zero-sum, see Appendix A.1 for proof). All values in-between, $b \in (0°, 180°)$, represent the spectrum of partially cooperative/competitive *general-sum* games. Figure 1a gives an instance of this game; the game's algorithm is given in Algorithm 1 in Appendix B. This can be seen as the game from Crawford & Sobel (1982) modified to cover the range of cooperative/competitive games.

### 3.1 TRAINING DETAILS

Both agents are implemented as MLPs with two hidden layers and ReLU (Nair & Hinton, 2010) nonlinearities between all layers. The targets are sampled from the circle, the sender takes its target, $T_s$, as input and outputs a categorical distribution over a vocabulary from which we sample a message—its output. The receiver takes the message as input and deterministically outputs its action, $a$. Errors are calculated using the $L_1$ loss on the circle. The sender estimates its loss using the score function estimator (Fu, 2006)—also known as REINFORCE (Williams, 1992)—and has

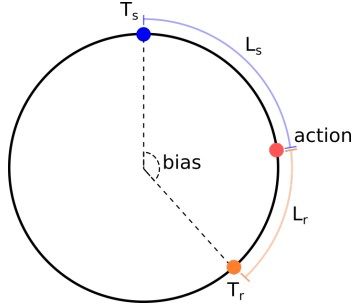

(a) The circular, biased sender-receiver game

Figure 1: Circular Biased Sender-Receiver Game has both agents given targets $T_R, T_S$ that are $b$ apart and choose an action $a$ to receive the $L_1$ losses $L_1^r, L_1^S$

an added entropy regularisation term. Since the loss is differentiable with respect to the receiver, it is trained directly with gradient descent, so we are training in the style of a stochastic computation graph (Schulman et al., 2015).

We train for 30 epochs of 250 batches, with batch size 64, and set the circumference of our circle to 36 (so that a loss of $90°$ is an error of 9). Both agents are trained using Adam (Kingma & Ba, 2014). To evaluate, we use a fixed test set of 100 equidistant points $\in [0°, 180°]$ and take the $\arg\max$ of output distributions instead of sampling. We do all hyperparameter searches with Oríon (Bouthillier et al., 2019), using random search with a fixed budget of (100) searches. We perform a hyperparameter search to over both agents' learning rates, hidden layer sizes, the vocabulary size, and entropy regularisation (when used). We always report results for given hyperparameters averaged over 5 random seeds, and we average our metric for hyperparameter search over the last 10 epochs to capture some level of stability as well as performance. All hyperparameter search spaces are available in the config files of the code repository.

## 4    COMMUNICATION: COOPERATION OR MANIPULATION

### 4.1    EVALUATING INFORMATION TRANSFER

To evaluate the communication emerged with a cheap-talk channel, we can simply look to the sum of agents' $L_1$ losses. Under non-communication (or uninformative communication), we know that the receiver will just guess a point at random, and the average loss for both players is the expected value of the loss—given that it is drawn uniformly $\mathbb{E}_{x \sim U(0,360)}[L_1^s(T_s, x)] = 90°$ Therefore, any error for either agent below $90°$ is evidence of information transfer (Lowe et al., 2019). Furthermore, since there is no other action space for agents to communicate in, the information transfer must be happening in the emergent communication space (Mordatch & Abbeel, 2017). Therefore, the lower $L_1^r + L_1^s$ is, the more informative the learned protocol is; and, the most informative protocol will have the lowest loss $\min_{a \in (0,360)} L_1^r + L_1^s = b$. To show this comparison, we always plot the loss under uninformative communication ($90°$) and the loss for each agent if they were to both fairly split the bias ($b/2$).

### 4.2    INFORMATION TRANSFER VS COMMUNICATION

While we have found evidence of information transfer, does that necessarily mean our agents have *learned to communicate*? For example, our hyperparameter search could find a minimal learning rate for the sender, such that it is essentially static, and a normal configuration for the receiver. The game would then become not one of learning a protocol between two agents, but rather just a receiver learning the sender's initial random mapping of targets to messages. The receiver could then dominate the sender by always choosing $a = T_r$, which would yield $L_1^r + L_1^s = b$; namely, the optimal sum of losses and, therefore, optimal information transfer. This situation is clearly not what we are looking for, but it would be permissible, or potentially even encouraged, under

an information-transfer objective (as measured by the sum of agents' $L_1$ losses). It is, therefore, necessary to delineate the differences in communication; here, we can look to extant results in signalling (Skyrms, 2010).

### 4.3 COOPERATION VS. MANIPULATION

One perspective on information transfer is that of *manipulation* of receivers by senders (Dawkins & Krebs, 1978) or vice-versa (Hinde, 1981); this manifests as the domination of one agent over the other. We note that these situations are modelled as *cue-reading* or *sensory manipulation*, respectively, and are distinct from *signalling*—i.e., *communication* (Barrett & Skyrms, 2017). Accordingly, communication requires *both* agents to receive a net benefit (Krebs & Dawkins, 1984), which implies some degree of *cooperation* (Lewis, 1969). For the fully-cooperative case, previous metrics of joint reward (Lowe et al., 2019), or even influence of communication (Jaques et al., 2018), are sufficient to drive the hyperparameter search. But for competitive scenarios, neither of these can distinguish between manipulation and cooperation (Skyrms & Barrett, 2018).

Since our focus is on the emergence of *cooperative communication*, we are looking for settings where both agents perform better than either their fully-exploited losses ($L_1^s < b$ and $L_1^r < b$) or the loss under non-communication ($L_1^s < 90°$ and $L_1^r < 90°$). With this goal in mind, we choose the sum of squared losses ($(L_2^s)^2 + (L_2^r)^2$) as our hyperparameter-search metric. We can view our partially competitive scenario as having a *common-interest loss* ($180° - b$), in which both agents are fully cooperative, and a *conflict-of-interest loss* ($b$), in which both agents are fully competitive. The sum of $L_1$ losses optimises only for the common interest, whereas $L_2$ prefers a more fair division of the conflict-of-interest loss in addition to optimising common interest (see proof in Appendix A.2). We use the $L_2$ metric only on hyperparameter search and keep $L_1$ as our game's loss to maintain a constant-sum game for the fully competitive case.

## 5 EMERGING COOPERATIVE SELFISH COMMUNICATION

### 5.1 COMMUNICATION IS PROPORTIONAL TO COOPERATION

We use six equidistant values of $b \in [0, 30°, 60°, 90°, 120°, 150°]$ and for each one, we do a hyperparameter search to find the lowest achievable $L_s^2 + L_r^2$. We do not usually test $b = 180°$ because the game is constant-sum and therefore trivially $L_1^s + L_1^r = 180°$, but for completeness you can see the results of a hyperparameter search with $b = 180°$ in Appendix C Figure 16. We report our results in Figure 2 and find that agents do learn to cooperatively communicate without any special learning rules contrary to current literature. We can see that the performance decreases proportionately to the bias, meaning the sender is less informative with messages, forcing the receiver to be less accurate in its own guesses. This matches the theoretical results of Crawford & Sobel (1982); information transfer with communication is inversely proportional to the conflict of interest. Plots for each $b$ are in shown in Appendix C Figure 6. For the curve, we still plot the $L_1$ losses to maintain consistency and to make clearer the comparison to the no-communication baseline and the optimal information transfer (common interest maximisation).

We find that our results are basically unchanged between the different hyperparameter metrics; a relatively fair and useful protocol is learned by the agents, but this deteriorates in more competitive scenarios. This is clear when comparing the stability and relative efficacy of protocols in $b = 30°, 60°$, shown in Figures 2b, 2c, and that of $b = 90°$ shown in Figure 2d. We can understand this through the lens of honest communication, which can be taken advantage of in highly competitive scenarios. If, for example, the sender communicates, with complete honesty, its own coordinates, then the receiver can take advantage of this and choose its location exactly so that $L_r = 0$ and $L_s = b$. Comparing this situation to non-communication ($L_r = L_s = 90°$), it is clear that even fully-exploited communication is a strictly dominant strategy for $b < 90°$ (i.e, when the game is *more cooperative than competitive*).

### 5.2 IMPROVING COMPETITIVE COMMUNICATION WITH LOLA-DICE

For more competitive cases, fully-exploitable communication is no longer dominant, and active communication now requires both agents to cautiously cooperate. To achieve this cooperation, we

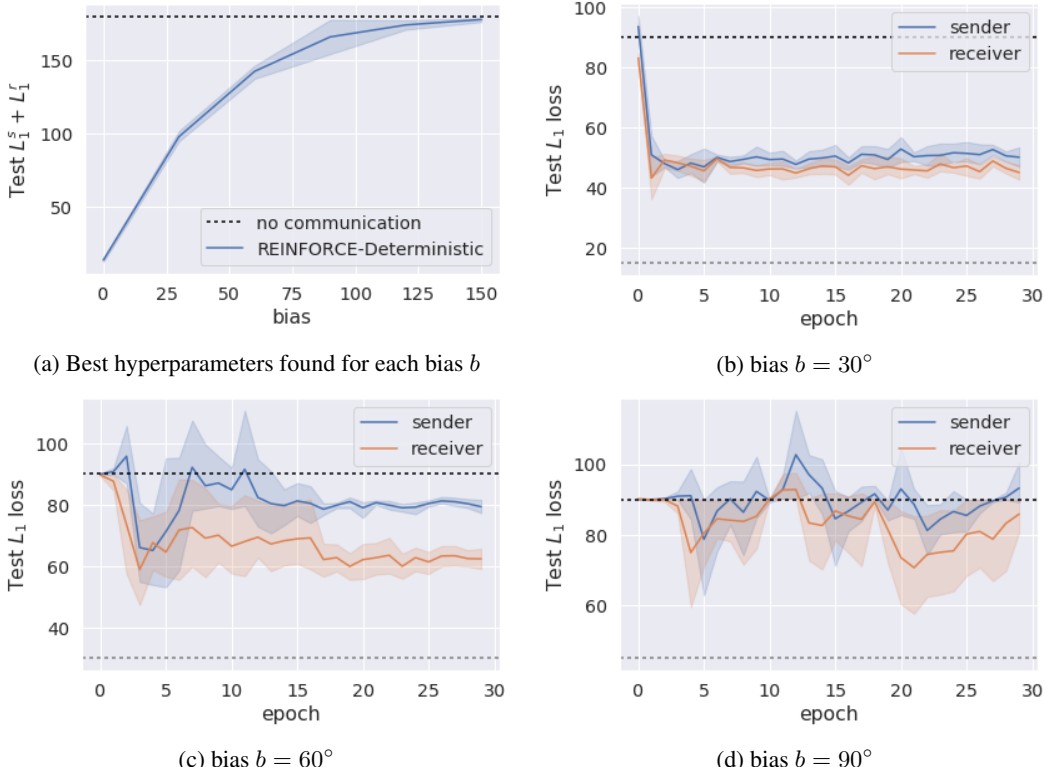

(a) Best hyperparameters found for each bias $b$

(b) bias $b = 30°$

(c) bias $b = 60°$

(d) bias $b = 90°$

Figure 2: All results are shown in Figure 2a we plot the lowest $L_1^r + L_1^s$ test loss found with a hyperparameter search for $b \in [0, 30°, 60°, 90°, 120°, 150°]$, demonstrating that informative communication (below the dashed line) is indeed learned by selfish agents. Note that performance even in the fully cooperative $b = 0$ is not optimal because of the bottleneck of discrete communication. For $b \in [30°, 60°, 90°]$ we show the training curve of the best hyperparameters found in 2b,2c,2d. We plot the test loss over training epochs and showing the mean and standard deviation over 5 seeds, finding that for $b < 90°$ we find stable and relatively fair communication is naturally learned

propose using LOLA (Foerster et al., 2018a)—a learning rule, resembling theory-of-mind, that allows us to backpropagate through $n$ steps of the opponent's *learning*. LOLA was able to emerge cooperative behaviour in an iterated prisoner's dilemma, so it is a prime candidate for resolving our game in a similar situation. We experiment with LOLA in three configurations—LOLA on the sender, LOLA on the receiver, LOLA on both—and do a similar hyperparameter search, with the added search space of the LOLA learning rate. Per the improvements made by Foerster et al. (2019), we replace the receiver's score function estimate with the DiCE estimator, and we backpropogate through exact copies of opponents as opposed to using opponent modelling. We show our results in Figure 3a with extended plots in Figures 8, 9, 10 in Appendix C.

We find that LOLA on the sender is ineffective, but LOLA on the receiver and on both agents does indeed lead to better performance. This implies that emerging communication in competitive scenarios necessitates cooperation and that this cooperation can be found through explicit opponent modelling. Furthermore, comparing the curves of basic agents (Figure 2d) with those of LOLA agents (Figure 3c) shows that gains in performance are not from one agent dominating the other, but from both agents improving and increasing stability. We also look at the performance of $n$-step LOLA, which backpropogates through $n > 1$ steps of opponent learning. Figure 3b demonstrates that 2-step LOLA slightly outperforms 1-step LOLA, but 3-step LOLA does not provide any increase over 2-step. We see from Figure 3d that the increase comes mostly from stability of learning and slight improvement on the part of the sender.

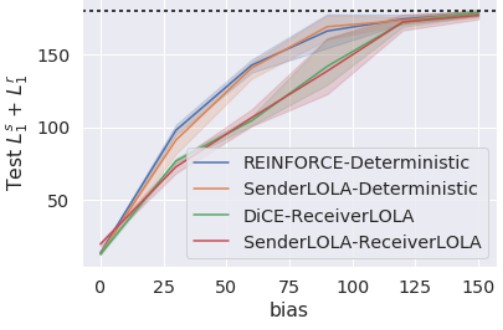
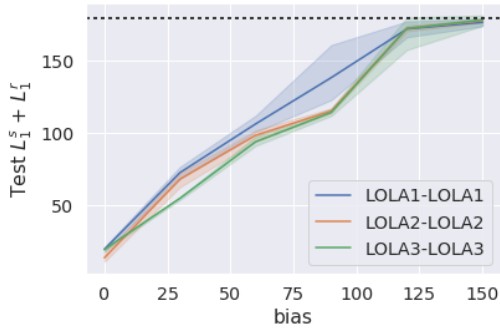

(a) Comparing the original setup, LOLA on the sender, LOLA on the receiver, and LOLA on both

(b) Comparing 1, 2, and 3-step LOLA on both agents

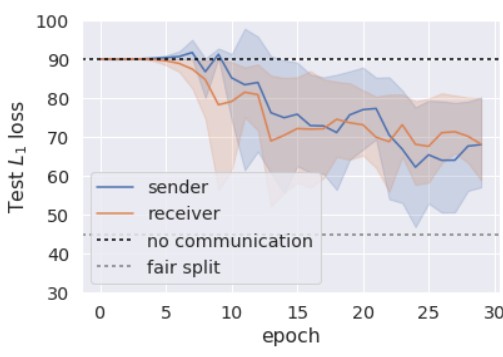
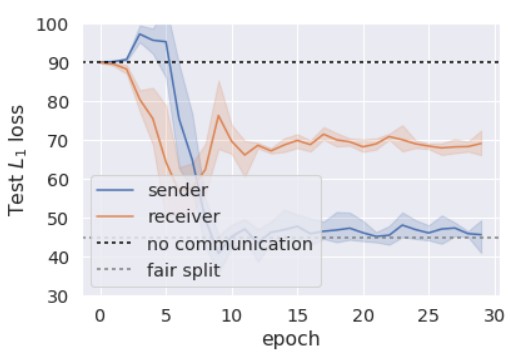

(c) 1-step LOLA on Sender and Receiver for $b = 90°$

(d) 2-step LOLA on Sender and Receiver for $b = 90°$

Figure 3: LOLA improves learning to communicate and it is especially visible at $b = 90°$ where our original setup does very poorly. 3b shows that higher step LOLA improves slightly further but not past 2-step. Best feasible communication protocols found for $b = 90°$ using 1-step (3c) and 2-step LOLA (3d) on both agents demonstrates that the gains in performance over the basic setup shown in Figure 2d are not just from one agent doing better (though the sender is doing better) but both agents improving in performance and stability. Shaded area is standard deviation over 5 seeds

## 5.3 DISCRETE VS CONTINUOUS COMMUNICATION

Another axis to consider is whether discrete or continuous communication lends itself better to learning with selfish agents. To compare, we make the sender's message a real-valued scalar and appropriately change its output distribution to be a Gaussian, for which it learns the mean and variance (concretely described in Algorithms 2 in Appendix B). We, again, run hyperparameter searches, and we consider training our baseline training with a REINFORCE Sender and deterministic receiver as well as training both agents with 1-step LOLA. Our results in Figure 4a suggest that the learned protocols for continuous communication are all highly informative and near optimal. However, in all cases, the receiver is learning to manipulate the sender, and there is little evidence of cooperative communication. Indeed, we found no cases of both agents having a net benefit ($L_r, L_s < 90°$) in *any* of the hyperparameter runs for continuous REINFORCE-deterministic agents past $b = 90°$, and we only two cases of net benefit for LOLA-1 agents. Comparing this to discrete communication with the same LOLA-1 agents in Figure 4f, we can clearly see that they have a preference for more cooperative behaviour. Thus, we find that discrete messages are an important component in emerging cooperative self-interested communication.

## 6 CONCLUSION

First and foremost, we show evidence against the current notion that selfish agents do not learn to communicate, and we hope our findings encourage more research into communication under

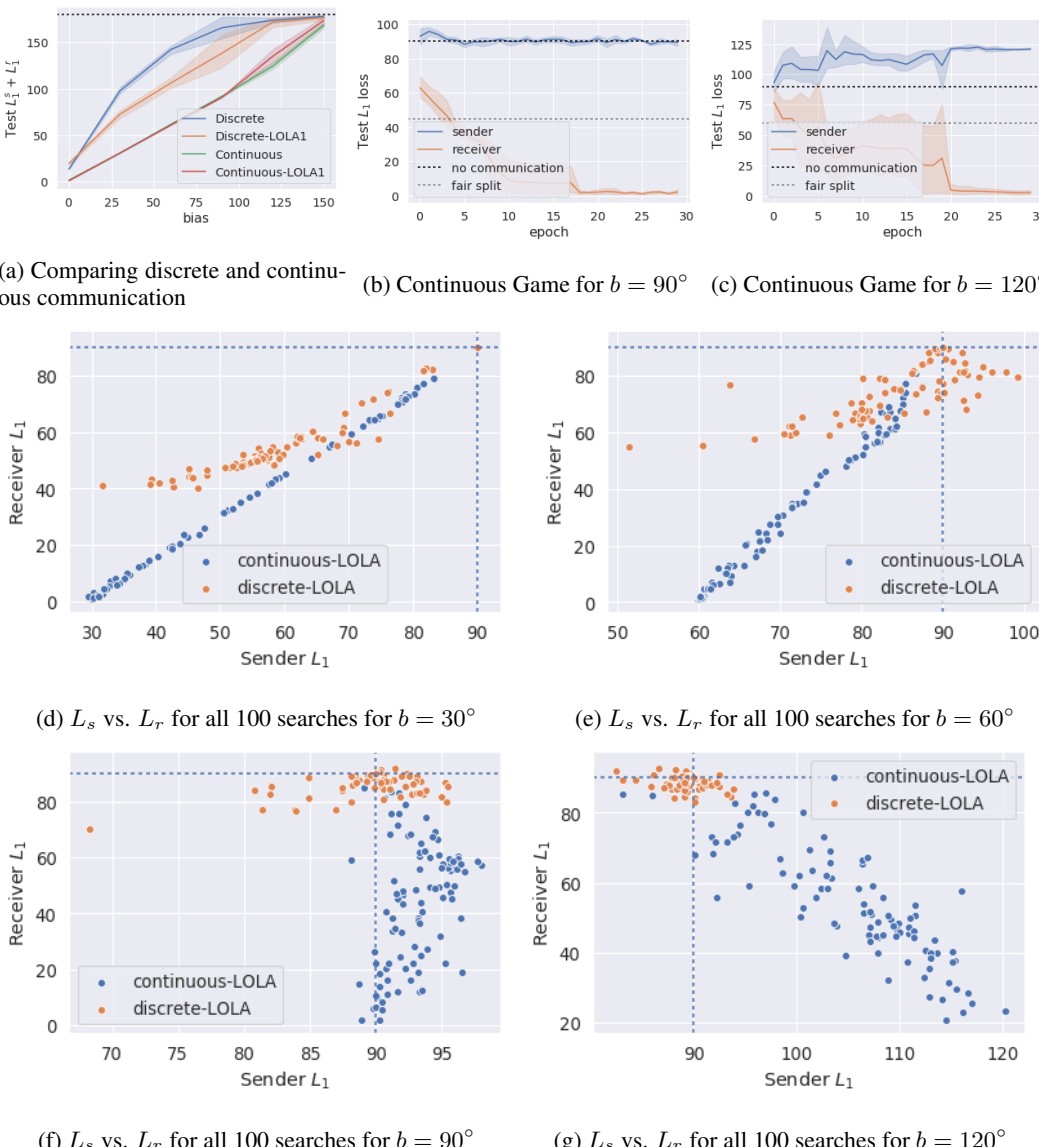

(a) Comparing discrete and continuous communication

(b) Continuous Game for $b = 90°$

(c) Continuous Game for $b = 120°$

(d) $L_s$ vs. $L_r$ for all 100 searches for $b = 30°$

(e) $L_s$ vs. $L_r$ for all 100 searches for $b = 60°$

(f) $L_s$ vs. $L_r$ for all 100 searches for $b = 90°$

(g) $L_s$ vs. $L_r$ for all 100 searches for $b = 120°$

Figure 4: The comparison between discrete and continuous communication for both the REINFORCE-deterministic setup as well as 1-step LOLA agents is shown in Figure 4a. We see that though overall continuous communication can achieve highest information transfer, the gains in performance seem to mostly from manipulation of the sender by the receiver. Two examples are shown for REINFORCE agents in Figures 4b,4c. To find a trend, we plot all 100 hyperparameter runs for $b \in [3, 6, 9, 12]$ between continuous and discrete communication using 1-step LOLA agents in Figures 4d,4e,4f,4g. We find that manipulation is the common result in continuous communication though individual cooperative points can sometimes be found. In general, continuous communication does not lend itself to cooperative communication

competition. We have shown three important properties of communication. First, a game being *more cooperative than competitive* is sufficient to naturally emerge communication. Second, we've clarified the distinction between information transfer, communication, and manipulation, providing motivation for a better quantitative metric to measure emergent communication in competitive environments. Next, we've found that *LOLA improves effective selfish communication* and, using our metric, we find it does so by improving both agents' performance and stability. Finally, we've shown that using a discrete communication channel encourages the learning of cooperative commu-

nication in contrast to the continuous communication channel setting, where we find little evidence of cooperation.

In fully-cooperative emergent communication, both agents fully trust each other, so cooperatively *learning a protocol* is mutually beneficial. In competitive MARL, the task is *using an existing protocol* (or action space) to compete with each other. However, selfish emergent communication combines these two since the inherent competitiveness of using the protocol to win is tempered by the inherent cooperativeness of learning it; without somewhat agreeing to meanings, agents cannot use those meanings to compete (Searcy & Nowicki, 2005; Skyrms & Barrett, 2018). Thus, the agents must both *learn* a protocol and *use* that protocol simultaneously. In this way, even while competing, selfish agents emerging a communication protocol must learn to cooperate.

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

## A  PROOFS

### A.1  PROOF OF FULLY COOPERATIVE/FULLY COMPETITIVE GAME

For $b = 0$, $T_s = T_r$ so trivially $L_s = L_r$ and the game is fully cooperative.

For $b = 180°$, $T_r = T_s + 180 \mod 360°$ we provide a visual demonstration in Figure 5 that the sum is always $L_s + L_r = 180°$ and therefore the game is constant-sum and fully competitive. We can also think of this as moving the actions distance $d$ towards one agent's target means moving it distance $d$ away from the other agent's target.

$0 \leq T_r, T_s, a \leq 360°$. Assume without loss of generality $T_s < T_r$ so $T_r = T_s + 180°$ and $T_s \leq 180° \leq T_r \leq 360°$

$$L_s + L_r = L_1(T_s, a) + L_1(T_r, a)$$
$$= \min(|T_s - a|, 360° - |T_s - a|) + \min(|T_r - a|, 360° - |T_r - a|)$$

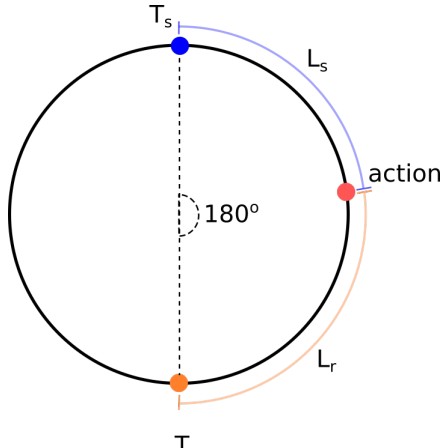

Figure 5: The game with maximal bias $180°$ showing the sum of $L_1$ losses $L_r + L_s = 180°$

**case 1**: $|T_s - a| < 360° - |T_s - a|$

$$\min(|T_s - a|, 360° - |T_s - a|) = |T_s - a|$$
$$|T_s - a| < 180°$$

**subcase a**: $T_s >= a$

$$|T_s - a| = T_s - a$$
$$\because T_r = T_s + 180°$$
$$T_r \geq a + 180°$$
$$\therefore \min(|T_r - a|, 360° - |T_r - a|) = 360 - (T_r - a)$$
$$L_s + L_r = T_s - a + 360° - (T_r - a)$$
$$= T_s - a + 360° - T_s - 180° + a$$
$$= 180$$

**subcase b**: $T_s < a$

$$|T_s - a| = a - T_s$$
$$a - T_s < 180°$$
$$a < T_s + 180°$$
$$a < T_r$$
$$\therefore |T_r - a| = T_r - a$$
$$\because T_r = T_s + 180°$$
$$T_r < a + 180°$$
$$T_r - a < 180°$$
$$2(T_r - a) < 360°$$
$$T_r - a < 360° - (T_r - a)$$
$$\therefore \min(|T_r - a|, 360° - |T_r - a|) = T_r - a$$
$$\therefore L_s + L_r = (a - T_s) + (T_r - a)$$
$$= T_r - T_s$$
$$= 180°$$

We can extend the proof by symmetry (on the circle) for $|T_s - a| \geq 360° - |T_s - a|$, so the sum of losses $L_r + L_s$ always equals $180°$ so the game is constant-sum and therefore fully competitive.

## A.2 Proof of $L_2$ Fairness

Assume without loss of generality $T_s < T_r$, we are minimizing the sum of $L_2$ losses

$$\min_a L_s + L_r = \min_a (T_s - a)^2 + (T_r - a)^2$$
$$= \min_a (T_s - a)^2 + (T_s + b - a)^2$$

let $T_s = x$

$$= \min_a (x - a)^2 + (x + b - a)^2$$
$$= \min_a x^2 - 2ax + a^2 + x^2 + 2bx + b^2 - 2ax - 2ab + a^2$$
$$= \min_a 2(x - 2ax + a^2 + bx - ab + b^2/4) + b^2/2$$
$$= \min_a 2(x - a + b/2)^2 + b^2/2$$
$$a = x + b/2$$

Sum of $L_2$ losses is minimized when the action is $T_s + b/2$ or halfway between both agents' targets.

## B   Algorithms

---
**Algorithm 1** Circular Biased Sender-Receiver Game

---
**procedure** Training Batch($b$)
   $T_s \sim \text{Uniform}(0, 360)$
   $T_r \leftarrow T_s + b$
   $m \sim \text{Categorical}(S(T_s))$
   $a \leftarrow R(m)$
   $L_s \leftarrow L_1(T_s, a) = \min(|T_s - a|, 360 - |T_s - a|)$
   $L_r \leftarrow L_1(T_r, a) = \min(|T_r - a|, 360 - |T_r - a|)$
   $R$ is updated with SGD
   $S$ is updated with REINFORCE or DiCE

---

---
**Algorithm 2** Continuous Circular Biased Sender-Receiver Game

---
**procedure** Training Batch($b$)
   $T_s \sim \text{Uniform}(0, 360)$
   $T_r \leftarrow T_s + b$
   $\mu, \sigma \leftarrow S(T_s)$
   $m \sim \text{Gaussian}(\mu, \sigma)$
   $a \leftarrow R(m)$
   $L_s \leftarrow L_1(T_s, a) = \min(|T_s - a|, 360 - |T_s - a|)$
   $L_r \leftarrow L_1(T_r, a) = \min(|T_r - a|, 360 - |T_r - a|)$
   $R$ is updated with SGD
   $S$ is updated with REINFORCE or DiCE

---

## C   Plots

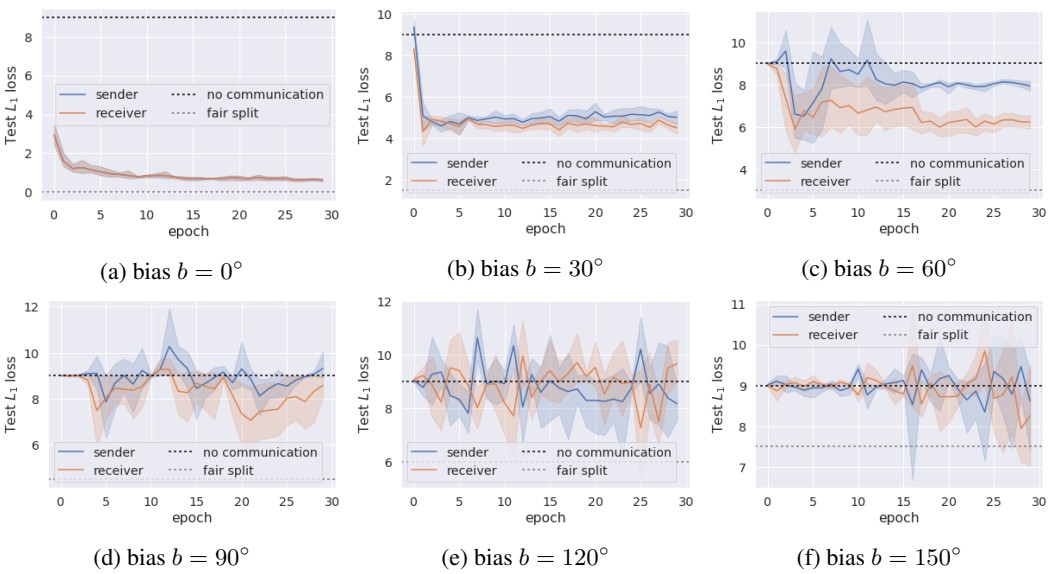

Figure 6: REINFORCE Sender, Deterministic Receiver, $L_1$ hyperparameter search

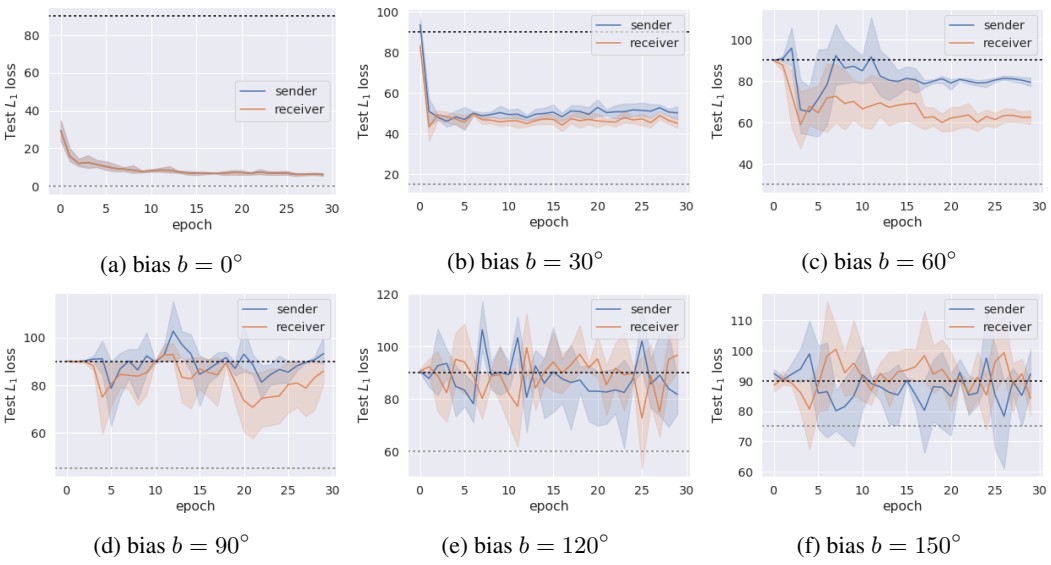

Figure 7: REINFORCE Sender, Deterministic Receiver, $L_2$ hyperparameter search. Note these are identical to Figure 6 except for $b = 150°$

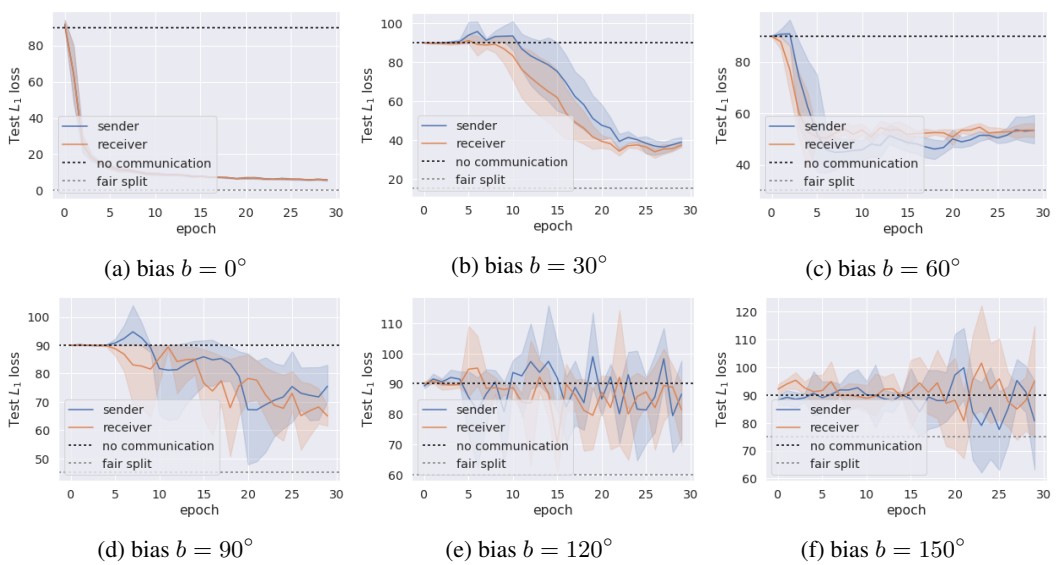

(a) bias $b = 0°$    (b) bias $b = 30°$    (c) bias $b = 60°$

(d) bias $b = 90°$    (e) bias $b = 120°$    (f) bias $b = 150°$

Figure 8: DiCE Sender, LOLA-1 Receiver

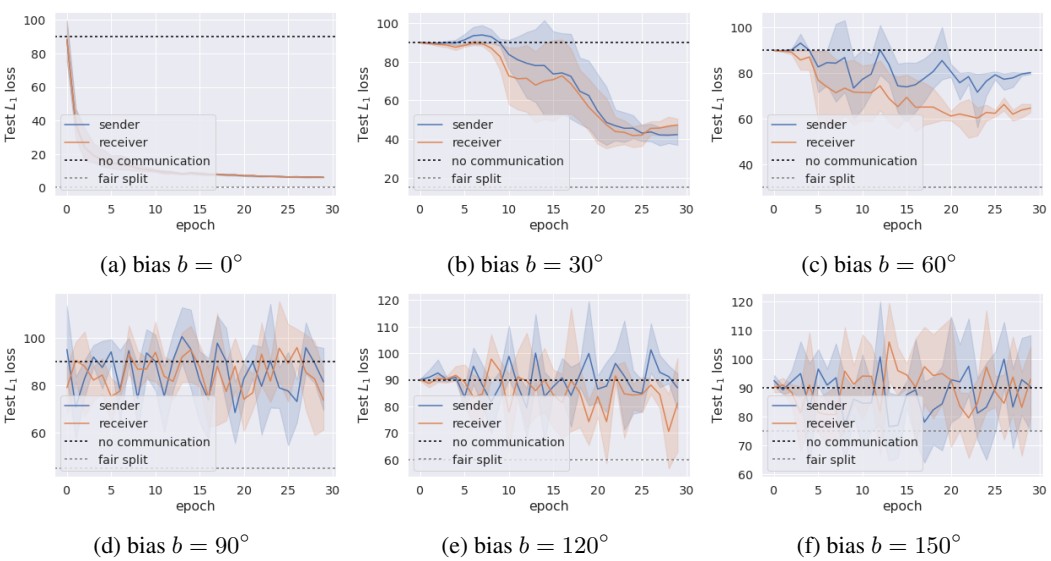

(a) bias $b = 0°$    (b) bias $b = 30°$    (c) bias $b = 60°$

(d) bias $b = 90°$    (e) bias $b = 120°$    (f) bias $b = 150°$

Figure 9: LOLA-1 Sender, Deterministic Receiver

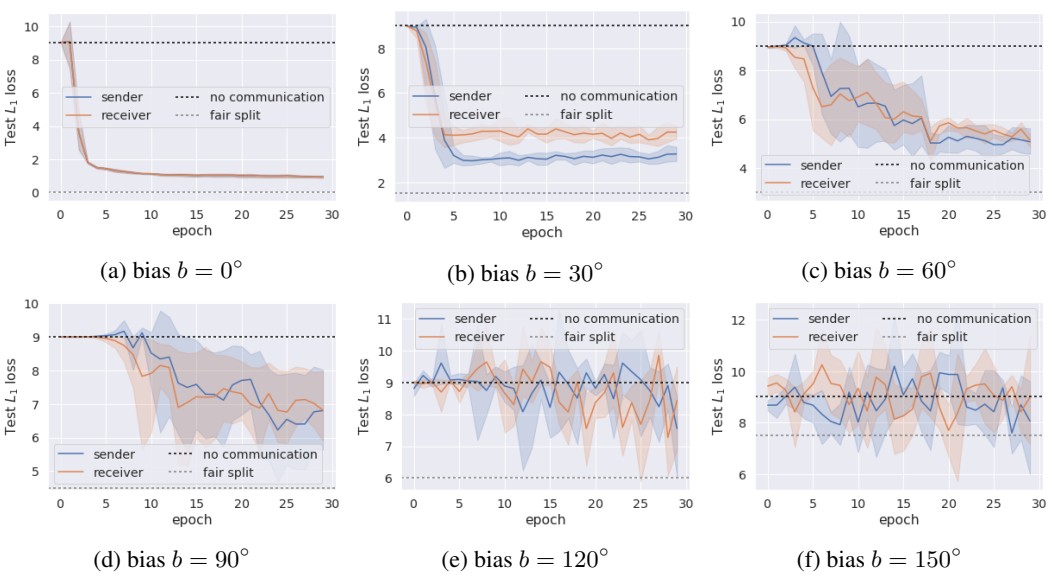

(a) bias $b = 0°$    (b) bias $b = 30°$    (c) bias $b = 60°$

(d) bias $b = 90°$    (e) bias $b = 120°$    (f) bias $b = 150°$

Figure 10: LOLA-1 Sender, LOLA-1 Receiver

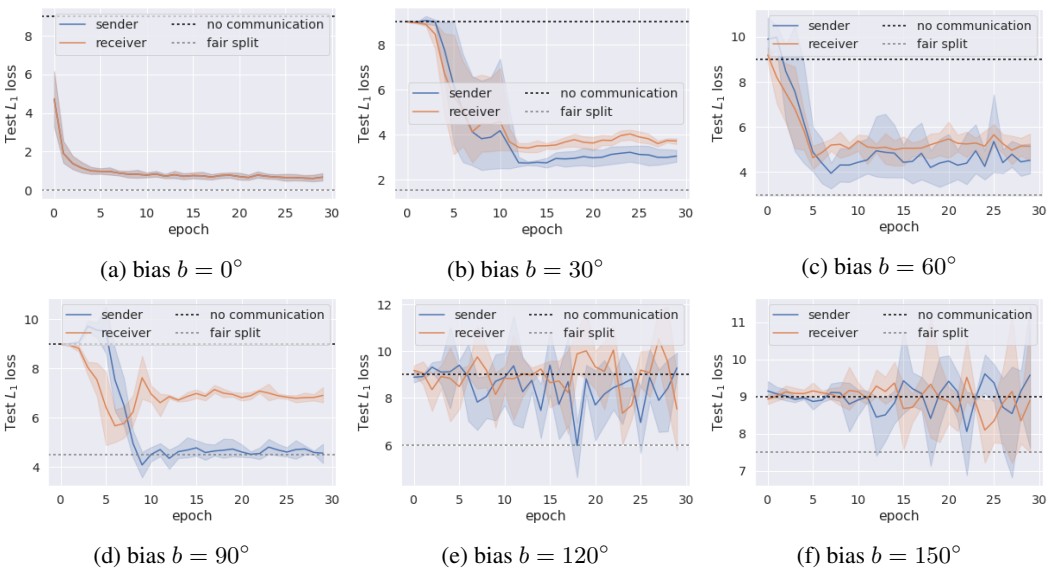

(a) bias $b = 0°$    (b) bias $b = 30°$    (c) bias $b = 60°$

(d) bias $b = 90°$    (e) bias $b = 120°$    (f) bias $b = 150°$

Figure 11: LOLA-2 Sender, LOLA-2 Receiver

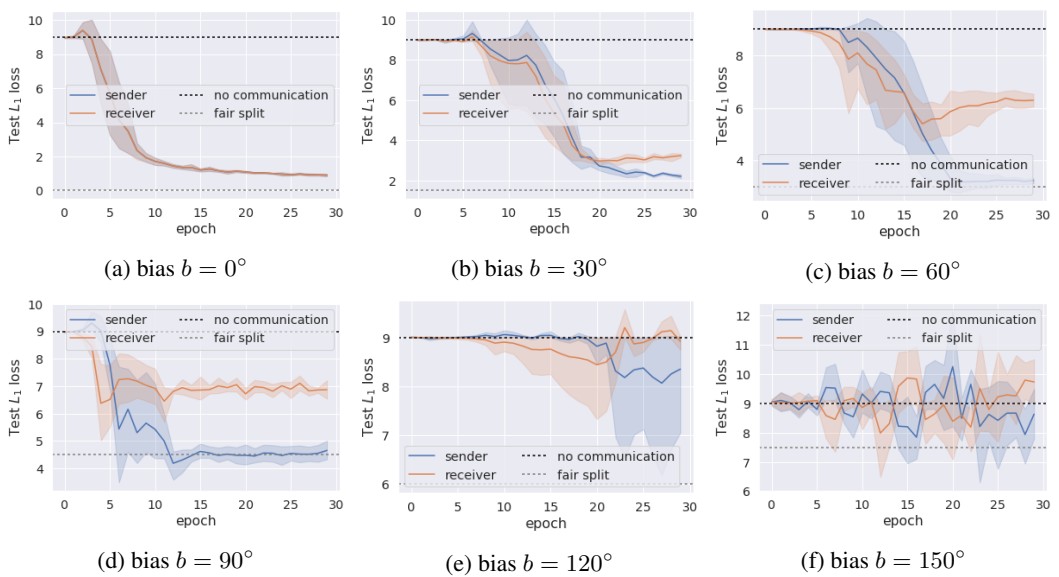

(a) bias $b = 0°$ (b) bias $b = 30°$ (c) bias $b = 60°$

(d) bias $b = 90°$ (e) bias $b = 120°$ (f) bias $b = 150°$

Figure 12: LOLA-3 Sender, LOLA-3 Receiver

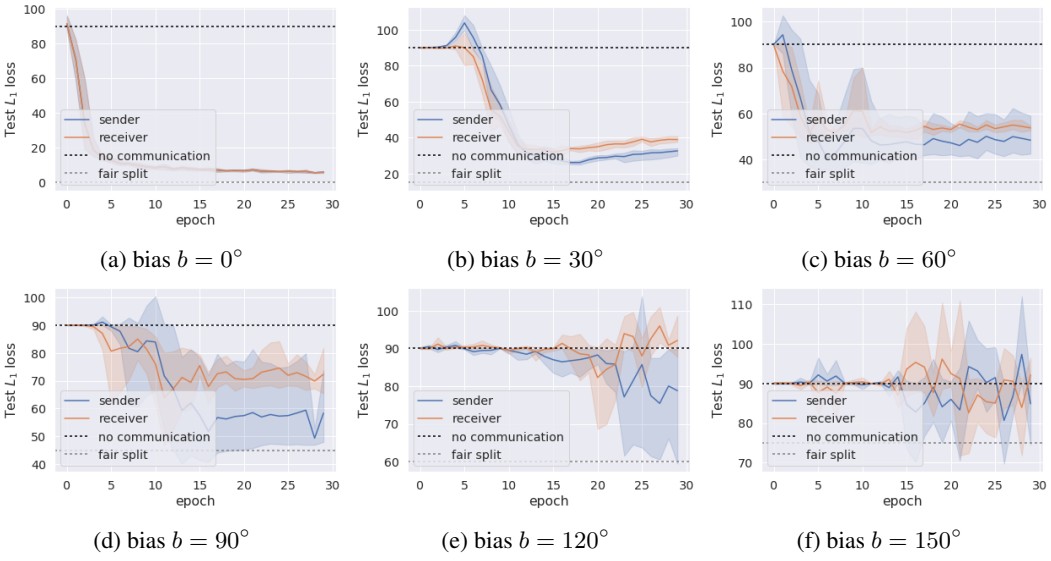

(a) bias $b = 0°$ (b) bias $b = 30°$ (c) bias $b = 60°$

(d) bias $b = 90°$ (e) bias $b = 120°$ (f) bias $b = 150°$

Figure 13: LOLA-4 Sender, LOLA-4 Receiver

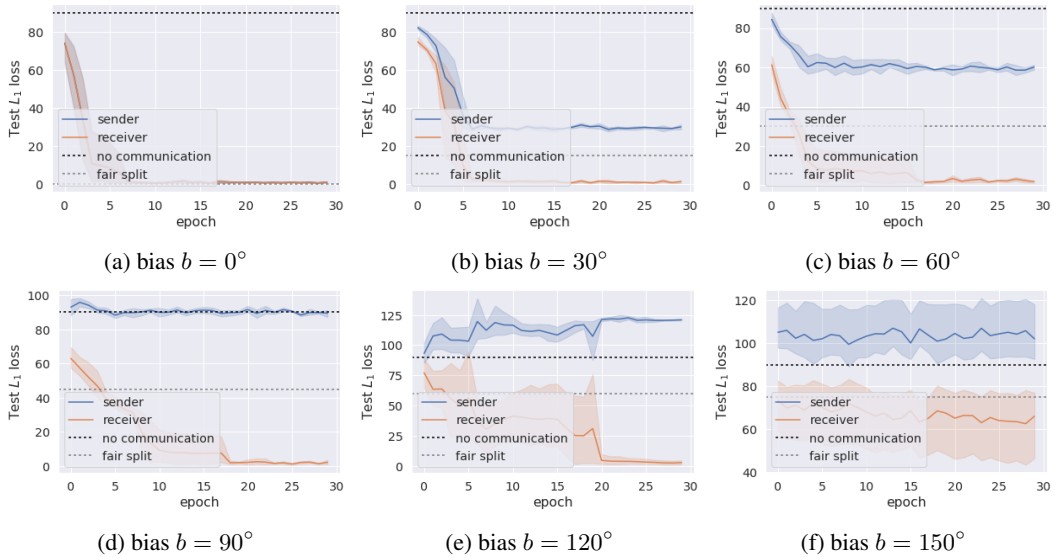

Figure 14: Gaussian Sender. Deterministic Receiver playing the continuous game

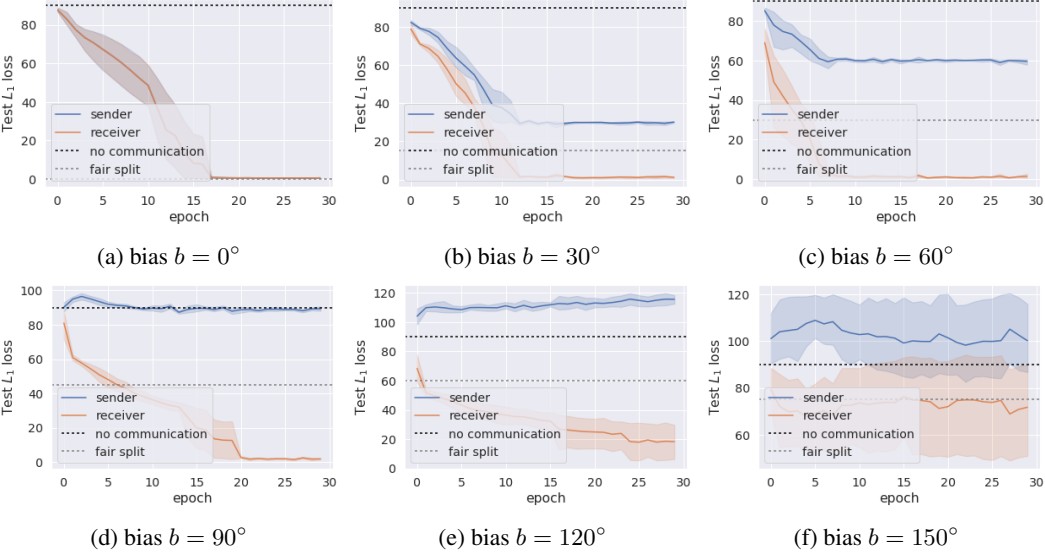

Figure 15: Gaussian LOLA Sender. LOLA Receiver playing the continuous game

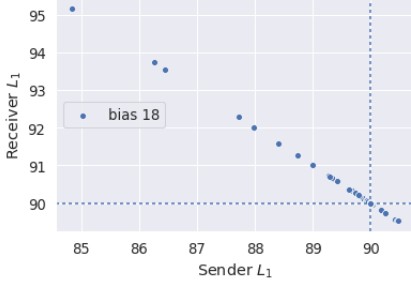

Figure 16: REINFORCE Sender vs Deterministic Receiver errors for all hyperparameter runs with $b = 180/degree$ shows that agents are mostly fair in the fully competitive constant-sum game

