# OpenReview forum: "Selfish Emergent Communication"
_ICLR.cc/2020/Conference — Reject_

### Official Review · AnonReviewer1 · 2019-10-24
**Official Blind Review #1**

**Rating:** 1

**Review:**

Summary:
This paper introduces a new sender-receiver game to study emergent communication in partially-competitive scenarios. The authors find that communication can also emerge in partially-competitive scenarios and demonstrate how to encourage communication: 1) selfish communication is proportional to cooperation, and it naturally occurs for situations that are more cooperative than competitive, 2) stability and performance are improved by using LOLA, and 3) discrete protocols are better than continuous ones.

Strengths:
- This is an interesting paper that is well written and motivated.
- They have justified a new sender-receiver game that can be tuned for various levels of competition which then allows them to analyze the effects of various levels of cooperation and competition.
- They perform sufficient experimental analysis to show that LOLA outperforms standard methods like REINFORCE in these settings and that discrete communication lends to cooperative communication.
- Evaluation is good in the sense that they repeat their experiments multiple times across different random seeds.

Weaknesses:
- Given that cheap talk is an extremely well-studied topic in economics, I feel that the authors should have devoted more time to explain the difference in setting between their work in classic pieces like those of Sobel and Crawford. The authors should properly define what they mean by learning agents versus fully rational agents, and the key differences between the two. Furthermore, nowhere do the authors in the cheap talk paper assert that agents use an existing language: the equilibrium itself assigns meaning to each sender’s message; this is not part of the problem definition per se. In fact, the work of Sobel and Crawford does not even constrain the size of the vocabulary (as was done in this paper): one of its key contributions is to show that in strictly non-cooperative settings, all equilibria must be partition equilibria, with only a finite number of messages used.
- Section 4.2: “initial random mapping of targets to messages.” The authors made the assumption that this mapping has to be deterministic. Absent a proof or a citation, I find this difficult to accept. This is especially so since mixed strategies are a crucial component of games of imperfect information.
- The introduction of the circular game is suspect. There already exist numerous games involving cheap talk, one of them from the Sobel and Crawford paper. Why is there a need for this new benchmark?
-  The description of the game is given as an algorithm in the appendix. This comes across as counterintuitive: why are the gradient steps being included as part of the game description? A game’s specification and the algorithm which is being used to solve it are two different things.
- It is difficult for me to assess the significance of these results since the authors have not presented real-world scenarios and experiments that demonstrate the importance of selfish communication. For cooperative communication we see it a lot in examples like grounded language learning, visual dialog, multi-agent communication etc. But I am concerned that the new setting proposed in this paper seems like a 'toy setting' to investigate if emergent communication would happen.
- Are the communicated symbols (discrete or continuous) semantically meaningful? It was shown in Kottur et al. (2017) that for emergent communication to occur and generalize to unseen test instances, it was crucial that the communication protocol was grounded i.e. one symbol learning to represent the color, one representing the shape, one representing the size. What is the final communication protocol learned in this case, and it is useful/interpretable in a similar sense?
- Typo: 'Figure ??' in line 3 of section 5.1

**Experience Assessment:**

I have read many papers in this area.

**Review Assessment: Checking Correctness Of Derivations And Theory:**

I assessed the sensibility of the derivations and theory.

**Review Assessment: Checking Correctness Of Experiments:**

I assessed the sensibility of the experiments.

**Review Assessment: Thoroughness In Paper Reading:**

I read the paper at least twice and used my best judgement in assessing the paper.

---

> ### Author Response · Authors · 2019-11-11
> **Response to Reviewer 1 Part 1**
>
> Thank you for your comments and corrections! We’ve taken quite some time to mull everything over and address your concerns point by point below. We would be happy to discuss further and more in depth.
>
> Crawford and Sobel
>
> - We put a discussion about the differences between our paper and Crawford and Sobel (1986) in a post above.
> - On your point about our phrase “an existing language”, we agree and are also happy to revise the wording. Our point was more subtle that they look at specific equilibria where there exists a fixed mutual understanding as opposed to looking at learning dynamics where the language is emerged and in flux. Would it be more appropriate to write they study “fixed languages at equilibria”?
>
> Section 4.2 Deterministic Mappings
>
> - In section 4.2, we do not make an assumption of deterministic mappings and indeed, during training, the sender is stochastic, choosing a symbol based on categorical distribution over a vocabulary (as is standard in emergent communication).
> - Our main point was that given a random initialization of a non-learning sender (learning rate close to 0) and a learning receiver with a regular learning rate, it is highly likely that the learning agent would dominate.
>     - This does not necessitate a deterministic sender since a stochastic sender’s mappings can be mostly learned (and therefore dominated) in nearly all cases.
>     - The only case where a non-learning sender cannot be dominated by a learning receiver would be a sender with a Nash policy (e.g. all states are mapped to the same symbol and communication is uninformative). But initializing a sender to a Nash policy is very unlikely given the random initialization methods of neural networks.
>     - So in the vast majority of cases, the learning agent would indeed do significantly better than its non-learning opponent.
> - We can revise the example to make it more clear that the situation is highly likely but not guaranteed.
>
> Justifying The Circular Game
>
> - We would like to clarify that this game is not a benchmark but closer to a diagnostic tool.
>     - We wanted to run experiments on the full range of 2 player cooperative/competitive games and empirically show that selfish emergent communication can be feasibly achieved
>     - We also wanted to demonstrate that it is indeed the bias of the game that influences the level of communication achieved and can explain why previous literature was mistaken.
> - We believe the game is really an extension of Crawford and Sobel, made to be smoothly tuneable from fully cooperative to fully competitive.
>     - To our knowledge, there does not exist such a game in game theory literature (Crawford and Sobel’s original game is not as easily made fully-competitive).
>     - To our knowledge, no existing games in emergent communication literature have fine-grained control of the level of cooperation/competition in the game.
>
> Algorithmic Game Description
>
> - We added the extra line about updates to make it more clear when the episode ends and agents can update their weights.
> - This is indeed not explicitly part of the game and we can remove it if it seems superfluous to understanding how the game is played by our agents.
>
> Realism of Selfish Communication
>
> - We would like to stress that emergent communication is a “how-possibly” explanation (Resnick, 1991) of language emergence.
>     - In this way, we think that reward-sharing and full cooperation is not as realistic of a model as two selfish agents that emergent communication for selfish reasons given an environment that requires cooperation.
>     - Current literature in emergent communication has usually assumed reward-sharing and therefore is less realistic than our setting.
>
> - On the topic of “toy setting”, this is indeed exactly a toy setting to see if communication emerges.
>     - Since literature in the field of emergent communication has implied communication does not emerge, we created this toy task to see if it could and whether Crawford and Sobel’s equilibria could be feasibly achieved in the modern setting of emergent communication.
>     - We do not make any presumptions about all games, nor do we think our game should be a benchmark. What we have shown is that research in emergent communication under competition should make use of strong baselines with selfish agents and take into account the quantifiable cooperation/competitive nature of the games studied, which it sometimes does not.
>
>
> --- continued below ---

---

> ### Author Response · Authors · 2019-11-11
> **Response to Reviewer 1 Part 2**
>
>
> Semantic Meaning Of Communication
>
> - For emergent communication to consistently improve performance over non-communication, it must be semantically meaningful.
>     - So our protocol is meaningful, but we are more concerned with how effective the meanings are as opposed to what they are exactly.
> - And communicative efficacy is better measured with rewards rather than qualitative methods (see Lowe et al 2019)
>     - Qualitative methods can be tricked by spurious mutual information metrics caused by certain network architectures
>     - Looking at reward with communication vs non-communication is a clear and foolproof way of measuring the efficacy
>
> - Could you clarify what you mean by “grounded”?
>     - We use “grounded” to refer to a symbol having a semantic meaning (essentially a mapping of symbols -> meanings). Any effective emergent communication can be said to be at least partially “grounded” since there must be semantic meaning conveyed by the agents in order to be effective.
>     - Kottur et al say that “compositional language is one of the optimal policies”  and point to the compositionality of grounded meanings as necessary for generalization
>     - For now, this remains a difficult term to encapsulate and we think the community has many meanings for it. Sidenote: Chris Manning had a great little speech on this exact point at last year’s ViGIL workshop (https://bluejeans.com/playback/s/jftkhICjhUnEbcglGD4qWWpHsvunBNISIZNdGdUo2AD7vD9nAq5aI2yXus70immP in Chapter 2 starting at 1:05:27)
>
> References
>
> Resnik, David B. “How-Possibly Explanations in Biology”. Acta Biotheoretica (1991) ,39(2):141–149.

---

### Official Review · AnonReviewer3 · 2019-10-24
**Official Blind Review #2**

**Rating:** 6

**Review:**

This ICLR submission deals with a problem of whether selfish agents can learn to use an emergent communication channel, using a sender-receiver game as a case study. It is found that communication can emerge in partially-competitive scenarios, and conditions in which this can happen are investigated.
This review is delivered with the caveat that I am not an expert in this particulat field.
The investigation seems relevant and the paper is well written and structured, being within the scope of the conference. Proofs in the appendix are sound to the best of my understanding.
The literature review is up to date and seems overall relevant.
This study should be understood as a proof of concept, given that the setting seems rather restrictive, so I am unsure that he results could be generalized.
They seem anyhow promising and partially challenge the current understanding of the problem.
Minor issues:
All acronyms in the text should be defined the first time they appear in the text.
LaTex problem with Fig. reference at the beginning of section 5.1.

**Experience Assessment:**

I do not know much about this area.

**Review Assessment: Checking Correctness Of Derivations And Theory:**

I assessed the sensibility of the derivations and theory.

**Review Assessment: Checking Correctness Of Experiments:**

I assessed the sensibility of the experiments.

**Review Assessment: Thoroughness In Paper Reading:**

I read the paper at least twice and used my best judgement in assessing the paper.

---

> ### Author Response · Authors · 2019-11-13
> **Response to Reviewer 3**
>
> Thank you for your review and comments, we appreciate your view and we hope that the paper was readable and enjoyable to someone who isn’t an expert in emergent communication.
>
> We will fix the figure reference and would like to know which acronyms you found unfamiliar that we can define them when they are first used.
>
> You make an important point about the generalizability of our results, our setup is indeed a simplified game that spans only the range of 2-player cooperative/competitive games and our learning is simplified by a simple state/action space. The reasoning for this is to be able to carefully control the learning dynamics:
> - tunable bias that perfectly reflects level of competition
> - no communication through a non-linguistic action space
> - clear way to differentiate between manipulation and communication
> - quantified optimal cooperative performance (maximizing cooperative reward)
>
> More complex 2-player environments should only differ in the complexity of learning the state, action, and game dynamics. But the feasibility of learning to communicate should still be relevant and our hope is to encourage strong baselines on selfish emergent communication. We also want researchers who don’t successfully achieve communication to investigate the underlying reasons, from competitiveness to issues with learning dynamics.
>
> For 3+ players, we expect game dynamics to be different and indeed it could be trivially reformulated to be zero-sum (Balduzzi et al, 2019) so that concept is not meaningful. There are competitive situations not covered by 2 players, e.g. even if 3 agents are fully competitive with each other, two of them could be incentivized to cooperate with one another in order to defeat the third. Still, we think our fundamental claim is generalizable from the simple 2-player cooperative/competitive nature to the general idea that communication should emerge naturally if cooperation is always preferred to non-cooperation (which we quantify).
>
> Our second two points should also be valid. LOLA should still be an essential tool to model opponents and allow communication even in cases where cooperation can be exploited. And discrete communication may still be preferable to continuous.
>
> Still, we would look forward to expanding on the research question of selfish communication to 3+ agents and more complex scenarios (e.g. ad-hoc communication using meta-learning) in future work

---

### Official Review · AnonReviewer2 · 2019-10-31
**Official Blind Review #2**

**Rating:** 3

**Review:**

This paper looks at the question of emergent communication amongst self-interested learning agents. The paper finds that "selfish" (ie. self-interested) agents can learn to communicate using a cheap talk channel as long as the objective is partially cooperative.
The paper makes states that this is is a novel finding that contradicts the previous understanding of emergent communication in the literature (side point: at least some of the papers referenced for this claim did not at all make the claim).

I believe there is a major miss-understanding here: As noted in the paper, self-interested agents can learn to communicate in settings in which the reward function is cooperative. Furthermore, it is also known that in 2 player zero-sum there is no incentive to learn a communication protocol.
This clearly shows that talking about whether or not "selfish" agents can learn to communicate only ever makes sense within the context of a specific game / reward structure.

With this in mind, the main finding, agents learn to somewhat communicate with each other in a simple toy setting, with more communication happening when the payouts are more cooperative, is not very interesting.

This doesn't mean that there isn't a good paper to be written here, in principle. Finding simple settings in which SOTA multi-agent learning "fails", ie. doesn't find Nash policies, understanding why it fails and then finding ways to mend things is generally a good research direction. However, this would require a few things which are currently lacking from the paper: (1) clear understanding of the Nash policies for the different reward settings (2) Implementation of SOTA methods for MARL which are appropriate for this setting (3) In depth analysis of learning successes and failures, ideally in settings which have previously been studied in literature (given how task-specific this analysis necessarily is).

Regarding 2: General sum games will generally have mixed-strategies as Nash equilibria (just think 'rock-paper-scissors'). With this in mind, using a deterministic policy for the receiver is inappropriate for making any claims about learning in general sum games.
Furthermore, it is well known that independent gradient descent (IGD) is not generally going to converge in general sum games (consider the loss functions X * Y and - X *Y or matching pennies). So looking at the outcome of IGD without checking for convergence means the results could be just about anything. Indeed, we don't have to go all the way to writing about emergent communication or complex "sequential social dilemma" to study this, those issues can easily be found in (iterated) matrix games.

This gets us to the second major point of the paper. To the authors' credit,  LOLA [1] has been shown to help with convergence in general sum settings and to lead to the emergence of cooperation and reciprocity in iterated games.

However, the key point for the ‘cooperation’ part is iterated. In a single shot setting (which is explored in this paper), there is simply no way for the agents to reciprocate with each other. So in short, I do not believe the authors' interpretation that agents learn to cooperate with each other because of LOLA, but I do believe that LOLA can help with the learning of mixed strategies (at least for the sender, given that the receiver is deterministic) and with stabilizing convergence. Lastly, the part of the experimental section is dominated by large error bars and graphs that are difficult to interpret.


Other points:
-"..but train agents to emerge their own." (and many other instances). AFAIK "to emerge something" is grammatically wrong (and also sounds really odd).
-"Since the loss is differentiable with respect to the receiver, it is trained directly with gradient descent, so we are training in the style of a stochastic computation graph (Schulman et al., 2015).". This is a weird statement. You don't need SCGs for training a supervised objective. Also, note that the loss is also differentiable with respect to the action of the 1st agent. It is trivial in this setting to compute the true expected return, if that is what you are after. Note my point above about deterministic policies
-"We perform a hyperparameter search to over both agents’" -> spurious "to"
-"We investigate a similar scenario but concern ourselves with learning agents as opposed to fully-rational agents that have full knowledge of the structure of the game, and we do not assume that agents use an existing language, but train agents to emerge their own" .This would be interesting, if the game was complex.
- L_1 vs L - these symbols are used inconsistently, with the subscript _1 sometimes being applied and sometimes not.
-"we can look to extant results" - s/extant/extent?
-"We use the L2 metric only on hyperparameter search and keep L1 as our game’s loss to maintain a constant-sum game for the fully competitive case." - A few points: (a) the game is not in general constant sum (b) By doing this hyperparameter search the evaluation is strongly biased towards 'fair' attributions. This seems highly problematic.
-"We report our results in Figure ??" -> Broken reference.
-"We do not test b = 180◦ because the game is constant-sum and therefore trivially Ls1 + Lr1 = 180◦." -> So? It would still be interesting to see what learning agents do in this setting.

[1]: "Learning with Opponent Learning Awareness", Foerster et al.

[update: I have updated the score based on the discussion with the authors]. While the paper lacks execution and conceptual clarity, I believe the game itself is interesting and could serve as a starting point for more thorough investigation.


**Experience Assessment:**

I have published in this field for several years.

**Review Assessment: Checking Correctness Of Derivations And Theory:**

N/A

**Review Assessment: Checking Correctness Of Experiments:**

I assessed the sensibility of the experiments.

**Review Assessment: Thoroughness In Paper Reading:**

I read the paper thoroughly.

---

> ### Author Response · Authors · 2019-11-11
> **Reviewer 2 Response Part 1**
>
> Thank you for the in-depth comments, corrections, and suggestions. We’ve tried to address all your concerns point by point below and would be happy to discuss further and more in-depth.
>
> Papers Claiming Selfish Communication Doesn’t Work
>
> - After reviewing all the papers we agree that (Foerster et al 2016) and (Lazaridou et al 2018) are bad citations for previous literature making this claim, many thanks for bringing this to our attention.
> - We still believe that the view of emergent communication being possible only in cooperative settings is prevalent in the literature and believe this is an important misunderstanding to address.
> - For Cao et al (2018), a main claim is that selfish agents cannot learn to effectively emerge communication whereas agents that share a reward function do.
>     - “Selfish agents do not appear to ground cheap talk”
>     - They conjecture in section 3.2 that this is because the game is not iterated but we show this is not necessary (more on this lower down in our comment)
>     - Instead, the game is likely too competitive and it is not necessary to share a reward function in order to communicate
> - Jaques et al (2019) reiterate the claim of Cao et al (2018) and claim that their learning rule allows for communication between competitive agents whereas regular methods do not, without explicitly quantifying the cooperative/competitive nature of their games.
>     - “The IC metrics demonstrate that baseline agents show almost no signs of coordinating behavior with communication, i.e. speakers saying A and listeners doing B consistently. This result is aligned with both theoretical results in cheap-talk literature (Crawford & Sobel, 1982), and recent empirical results in MARL (e.g. Foerster et al. (2016);Lazaridou et al. (2018); Cao et al. (2018)).”
> - We also found that Lanctot et al (2017) imply that emergent communication is a purely cooperative task (in the sense that they take communication to be a paradigm of cooperation):
>     - “In MARL, several agents interact and learn in an environment simultaneously, either competitively such as in Go [92] and Poker [39,106,73], cooperatively such as when learning to communicate [23, 94, 36], or some mix of the two [59, 96, 35].”
>
> "Reward Function Is Cooperative"
>
> - We would like to make a small distinction between the reward function being cooperative and the game encouraging cooperation.
>     - In previous work in emergent communication, there have been papers that have simply given the same reward to both agents or given a part of one agent’s reward to the other explicitly (Lerer and Peysakovich 2018) which is a cooperative reward function.
>     - The reward function for our agents is purely their own — selfish. They do not, a priori, have cooperative intentions. It is only through discovering the nature of the current game’s setup that they should realize cooperation is advantageous.
> - Cao et al (2018) study a cooperative “prosocial” reward function in a game that is quite competitive whereas we study purely selfish reward functions but in a game whose competitive nature can be tuned by the bias.
>     - Regardless of the game “prosocial” agents are going to be cooperative
>     - We take the view of Shoham et al (2003) that if agents are not being controlled by a central designer then the interesting scenario is when “learning takes place by self-interested agents”, as opposed to prosocially-interested agents
>
> Misunderstanding
>
> - We could not figure out what is the exact misunderstanding you are pointing to. Could you rephrase it perhaps?
>
> "Uninteresting"
>
> - Though the result may seem uninteresting from the perspective of static analysis where Crawford and Sobel’s result is clear, we perform a dynamic analysis (this is explained in more detail in our Crawford and Sobel discussion).
> - We believe it is, at minimum, interesting for the emergent communication community that seems to hold an opposing belief.
>     - Since there are two well-cited publications from top conferences (ICLR, ICML) that clearly and unambiguously state that selfish agents do not learn to communicate and one implying emergent communication is purely cooperative, we believe that there is indeed a misconception of selfish emergent communication in the field that deserves to be clarified
>     - We believe that our toy task is sufficient to show that selfish emergent communication should be feasible to achieve with modern deep RL methods, overturning that belief
>
> -- continued below --

---

> > ### Comment · AnonReviewer2 · 2019-11-11
> > **reply to "part 1"**
> >
> > [Side note: I believe breaking the response into 4 parts goes against the spirit of the 5000 characters limit.]
> >
> > @(Foerster et al 2016) and (Lazaridou et al 2018) are bad citations for previous literature making this claim:
> > No problem.
> >
> >
> > @"- We could not figure out what is the exact misunderstanding you are pointing to. Could you rephrase it perhaps?"
> >
> > Yes - talking about whether or not 'self-interested' agents cooperate / communicate can only ever be meaningfully discussed given a specific game or reward structure. The claim of Cao et al (2018) should be seen in the context of their specific game.  Clearly, self-interested agents will learn to communicate in games where their payouts happen to be correlated, with a game of identical payouts being an extreme example.
> >
> > Along the same line: "- We would like to make a small distinction between the reward function being cooperative and the game encouraging cooperation.
> >     - In previous work in emergent communication, there have been papers that have simply given the same reward to both agents or given a part of one agent’s reward to the other explicitly (Lerer and Peysakovich 2018) which is a cooperative reward function. "
> >
> > It doesn't matter if you explicitly define a new reward function that is R1' = R1 + alpha * R2 or just come up with a game that is inherently cooperative within some range, where R1 happens to be R1' . The resulting problem is the same, so I don't think this distinction is meaningful.
> >
> > So to summarize: Self-interested (or 'selfish' as you like to call it) by itself is a meaningless distinction outside of a specific game or reward structure, since it includes the limiting case of two agents that are both self-interested in optimizing the same reward function.

---

> > > ### Author Response · Authors · 2019-11-13
> > > **Response To Misunderstanding + Reward Sharing**
> > >
> > > Thank you for getting back so promptly! Sorry about the long response, we didn’t initially realize the 5000 character limit and were just trying to be thorough. Thank you for going through it all and responding, we really appreciate it.
> > >
> > > Misunderstanding
> > > - We agree that self-interested agents should only be discussed within the context of a game
> > > - We don’t think Cao et al’s comments were meant to be just within the context of their game
> > >     - Jaques et al (2019) is from a similar set of authors and cites Cao et al for their point that Jaquest et al’s self-interested agents should not learn in a completely different game!
> > > - We believe our experiments clarify some issues in Cao et al (2018)
> > >     - Cao et al conjecture that the issue is that the game is not iterated. We show in our experiments that iteration is not necessary to emerge communication under competition which could be counter to their narrative
> > >     - The real reason communication does not emerge is likely two-fold 1. The game dynamics allow one agent to dominate under non-communication 2. The game is likely too competitive (their setup does not control the level of competition but our experiments found it to be generally high)
> > >     - We’ve investigated the negotiation game and gone into more detail in our reply above
> > >
> > > Cooperative Reward Function vs Specific Game Structure
> > > - There is a fundamental difference, specifically in MARL, between a game structured for cooperation and reward sharing: Reward sharing doesn’t guarantee cooperation on it’s own.
> > > - Take Cao et al and assume their negotiation game is zero-sum (it isn't always, based on the description). What does it mean for two agents playing a zero-sum game to be 30% competitive?
> > > - Let’s think about being competitive over 30% of the possible reward. Can we define 30% competitive with reward-sharing?
> > > - We can give both agents a reward $R  = 0.7 * R_{you} + 0.3 * R_{them}$ but what does that do? Given your weight for an item $W_you$ and your opponent's weight for it $W_{other}$ there's three cases:
> > > 1. both agree you should take the item ($W_{you} * 0.43 > W_{them}$ )
> > > 2. both agree they should have the item ($W_{them} * 0.43 > W_{you}$)
> > > 3. you compete over the item ($0.43 * W_{you} < W_{them} < 2.3 * W_{you}$)
> > > - Is this 30% competitive? It depends on the weights given to a player and the distributions of items being negotiated over.
> > > - Just doing reward sharing does not guarantee a specific level of competition. Even a partially cooperative reward function may not guarantee cooperation as the optimal strategy
> > > - Our point is that reward-sharing still needs a carefully constructed game to specify the level of competition and encourage cooperation, so we make a distinction between reward-sharing and actually guaranteeing the level of competition by carefully creating a game with cooperative/competitive dynamics.

---

> > > > ### Comment · AnonReviewer2 · 2019-11-14
> > > > **Reviewer response**
> > > >
> > > > @zero-sum:
> > > > See my comment above. A two-player negotiation game should not be zero-sum (otherwise there is no point in negotiating).
> > > >
> > > > @Reward sharing vs game design:
> > > > My claim was that there is no set distinction, that's all. In particular relying only on game design rather than reward sharing is by no means a restrictive assumption, since any reward sharing scheme could be simply implemented as a new 'selfish' game with updated rewards.

---

> > > > > ### Author Response · Authors · 2019-11-14
> > > > > **Author Response**
> > > > >
> > > > > Sorry, we meant "fully competitive" not "zero-sum", the argument is based on that assumption
> > > > >
> > > > > It seems we're agreed that a carefully constructed game is necessary (and one of the important ways we improve upon previous emergent communication work). The distinction between reward sharing vs selfish is secondary and we think selfish agents just make understanding the game/reward clearer, though functionally we could change the structure to reward-sharing.

---

> ### Author Response · Authors · 2019-11-11
> **Reviewer 2 Response Part 2**
>
> Nash Equilibria and MARL
>
> - Though studying equilibria and Nash may be useful, we believe that it is not something our paper on multi-agent learning should focus on. Our view is strongly influenced by Shoham et al (2003) which has informed modern deep MARL and we give a brief summary of related points here
>     - We are studying how agents “should” learn and therefore how agents “should” act
>     - One possible agenda is “equilibrium” and asks whether a vector of learning strategies forms an equilibrium
>     - Another possible agenda is “AI” and asks what is the best learning strategy given a fixed class of possible opponents
>     - The main difference between the two agendas is “bounded rationality”
>     - “Equilibrium” assumes perfect reasoning and infinite mutual modelling
>     - “AI” starts from a base of bounded rationality and only adds mutual modelling when necessary
>     - These two agendas are not necessarily mutually exclusive but there is distinct philosophical difference
> - Why our situation is better represented by the “AI” agenda and therefore should not be focused on convergence or Nash equilibria
>     - We believe the “bounded rationality” assumption to be more appropriate for language emergence which could be said to give a “how-possibly” model (Resnick, 1991) of the emergence of human language
>     - We already model our fixed class of opponents as being SGD learning models with similar loss structures. This class of opponents is an assumption made by LOLA and we believe it is reasonable for deep MARL.
>
>
> SOTA MARL
>
> - We believe that, though simple, our methods are indeed state of the art for the specific problem they are tackling.
>     - We do not know of better complex gradient estimators being used in such low-dimensional input situations
>     - We believe that our results are quite good and relatively close to the theoretical optimum
> - Do you have specific MARL learning algorithms you believe would perform better for our setup? We would be glad to implement and test them
>
> Analysis of Failures
>
> - We believe we do a fair analysis of successes and failures.
>     - We find the issue that likely was underlying why previous works did not emerge communication with selfish agents (competitiveness) and do a careful analysis to show how it could be possible
>     - We look at two different popular ways of emergent communication (continuous vs discrete) and analyse how they affect our situation and the achievability of good selfish emergent communication
>
> Deterministic Receiver
>
> - Deterministic receivers are standard in basic emergent communication (see examples in EGG by Kharitonov et al)
> - We could change our receiver to be stochastic, for example, by making its output a gaussian distribution over actions and then sampling from that. This would not make things too different.
>     - Since we train with gradient descent, we could use the reparametrization trick to get the gradient for the receiver (Kingma and Welling, 2014)
>     - Doing this would make our stochastic receiver no different from a deterministic receiver that has an added gaussian noise in its output
>     - This would essentially only be adding variance to the learning of our receiver and simply be worse from an optimization perspective without being too functionally different
>
> IGD and Convergence
>
> - We would like to clarify that we are not making theoretical arguments about all general-sum games
>     - Theoretically, Singh et al (2013) have shown that for 2-player 2-action general-sum games, independent gradient ascent with an infinitesimally small step size will lead to either convergence at a Nash equilibrium or an expected payoff equivalent to the payoff at some Nash equilibrium
> - Please see our point about Nash and MARL for a discussion on why we are not specifically looking for equilibria or convergence
>
> --- continued below ---

---

> > ### Comment · AnonReviewer2 · 2019-11-11
> > **response to Part 2**
> >
> > @Nash:
> > I strongly disagree. The one big advantage of a toy problem is that it can be studied in terms of equilibria, shedding light onto the learning.
> >
> > @"Deterministic receivers are standard in basic emergent communication (see examples in EGG by Kharitonov et al)":
> > This makes sense in fully cooperative settings, but not in general sum. As I pointed out in my review, even something as simple as rock-paper-scissors requires a mixed strategy.
> >
> > @SOTA marl:
> > Yes, I would start with a stochastic policy and an algorithm that actually has convergence guarantees in general-sum. Examples that come to mind at SGA (https://arxiv.org/abs/1802.05642) or SOS (https://arxiv.org/abs/1811.08469).
> >
> > @2-player 2-action general-sum games,.. or an expected payoff equivalent to the payoff at some Nash equilibrium:
> >
> > That is not the game you are playing. Also, I believe their statements hold for stochastic policies, not for deterministic ones. Lastly, in your paper you do not at all analyze averaged policies.
> >
> > @Analysis of Failures
> > I disagree. Your paper cannot analyze learning failures since you do not have an understanding of what the best case learning even is. This goes back to understanding the Nash equilibria of the game.
> > For example, even when the interests are partially aligned the agents stop learning to communicate. Is that happening because it's no longer an equilibrium or because learning is breaking down?

---

> > > ### Author Response · Authors · 2019-11-15
> > > **Author Response**
> > >
> > > We don't believe Nash equilibria are essential to our specific research but we do think they are useful and future work with a Nash analysis could extend and improve this current work
> > >
> > > Knowing nash equilibria does not give full clarity on learning dynamics
> > > - A Nash equilibrium is just a local attractor which does not guarantee clarity about general learning dynamics. Neural networks are famous for guarantees at possible minima while explanations of learning dynamics are still an active research area
> > > - Even if we found a learning algorithm that converged towards nash equilibria it could trivially find the non-communication equilibrium, which is the lower bound on performance anyways
> > > - “Nash equilibrium strategy has no prescriptive force. At best the equilibrium identifies conditions under which learning can or should stop (more on this below) ,but it does not purport to say anything prior to that” (Shoham et al, 2003)
> > >
> > > Nash equilibria can be useful for playing against new opponents but this is not possible without an alternative paradigm (e.g. meta-learning) for emergent communication
> > > - Agents co-learn a protocol and two agents trained separately cannot communicate with each other at test time
> > > - Even if an agent learned a Nash strategy, it would not be useful against another a new opponent who would not understand their language
> > >
> > > Finding and defining Nash equilibria are not necessary for any of our findings
> > > - Empirically showing that communication with selfish agents is possible and conditional on the cooperative nature of the game does not require knowing Nash equilibria
> > > - We know the upper bound on the effectiveness of communication (total error = bias) without needing to know whether there is a Nash equilibria there
> > > - Showing that the regular emergent communication setup fails to learn in competitive scenarios does not require Nash equilibria. Proving or disproving possible equilibria in the competitive failure cases would not definitively prove or disprove whether agents theoretically could learn to communicate (as they could learn a non-equilibrium or an unstable equilibrium)
> > > - LOLA is shown to improve cooperation and communication statistically significantly without needing to show whether it converges to equilibria
> > > - Our analysis of discrete/continuous communication is a purely practical and deals with learning dynamics
> > >
> > > What knowing Nash equilibria could help with
> > > - Setting tighter upper bounds on the achievable stable communication. Perhaps the communication found by LOLA is even closer to optimal
> > > - Stronger arguments on whether communication is feasible to achieve in highly competitive scenarios. A lack of Nash equilibria does not guarantee infeasibility but it is a possible indicator
> > > - Proof of stable competitive communication protocols
> > >
> > > Deterministic Receiver
> > > We noted your review but we point out the futility of making the receiver stochastic.
> > >     - If we implement a stochastic receiver as described, it would be functionally equivalent to a deterministic receiver. The difference is only adding noise during training.
> > > Rock paper scissors is a normal form game, whereas our game is extended form and therefore the second player is conditioned on the first and can be deterministic
> > >
> > >
> > > SOTA MARL
> > > The convergence guarantees for those two algorithms are only for differentiable games. Emergent communication with discrete messages is not a differentiable game
> > > SOS is an extension of LOLA that only improves performance in situations such as the tandem game.
> > > - In IPD, SOS was shown to be nearly identical to LOLA.
> > > - If our situation resembled the tandem game, LOLA would not improve upon regular RL
> > > - Since situation does not resemble the tandem game, we should not expect SOS to do any better than LOLA
> > > SGA
> > > - requires knowing the loss and jacobian of other players
> > > - converges to a Nash equilibria, which could be trivially non-communication
> > >
> > > Analysis of Failures
> > > Best case learning is clearly defined: a sum of players $L_1$ errors = the bias and one agent is not manipulating the other. This is not the best **stable** configuration but why does it need to be stable? Agents can maintain a reward with a communication protocol in flux
> > > We're not actually at equilibria previously so the issue with more competitive scenarios must be that learning dynamics break down (which is why LOLA, with better dynamics, can help)

---

> ### Author Response · Authors · 2019-11-11
> **Reviewer 2 Response Part 3**
>
>
> Iterated vs One-shot Game
>
> - This is a very interesting point and we are happy you brought this up
> - Though it seems like a simple one-shot game, two things seem to make cooperation possible here
>     1. A two stage game (1. Sender sends message 2. Receiver takes action). We did experiments on iterated prisoner’s dilemma (a one-stage game) and found that LOLA could not emerge cooperation in the one-round case though it does in the iterated game.
>     2. “Iteration in the parameter space”. Though the game itself is not iterated, the fact an agent plays with the same opponent throughout training allows them to learn conventions with their opponent e.g. it is trivial to learn a simple coordination game between RL agents that are trained together
> - LOLA is indeed the reason for improved cooperation and communication efficacy
>     - Communication at bias = 90 is clearly better with LOLA agents than it is with our basic setup
>     - It is specifically LOLA that is helping cooperation as making that one change is sufficient to get our results
>     - You can review our code, the only difference between the LOLA scenario and REINFORCE scenario is that specific agent’s loss function reflects LOLA updates, all other code is unchanged
>
> Confusing Graphs and Error Bars
>
> - Could you please specify which graphs are confusing and which error bars you feel are too large? We deeply care about the clarity and statistical significance of our paper and would be happy to improve it
>
> Sender's Differentiable Loss?
>
> - In "note that the loss is also differentiable with respect to the action of the 1st agent" we took "1st agent" to refer to the sender
> - The loss is not differentiable wrt to the sender
>     - Sender maps its state to a categorical distribution over symbols from a vocabulary and stochastically chooses a single symbol based on the distribution. This symbol is the message
>     - Receiver takes the message and deterministically chooses the target
>     - The loss is differentiable wrt to the message but not wrt to the parameters of the sender because of the stochastic choice
> - For this reason we need to use a gradient estimator
>     - For the basic setup we use REINFORCE with a mean baseline for variance reduction. This is standard in emergent communication (see EGG by Kharitonov et al, 2019)
>     - For the LOLA setup we use DiCE because it allows us to do higher order gradient estimation
> - This setup of a differentiable receiver and gradient estimated sender is an SCG
>     - It differs by having two different objectives, one for the sender and one for the receiver
>
> "This would be interesting if the game was complex"
>
> - We did not aim to find a difficult existing game and show that communication under competition could be achieved because
>     - Jaques et al (2019) already emerge communication in a more complex game, albeit with a complex learning rule
>     - And if we did do this for an even more complex game without being able to easily control the exact levels of competition/cooperation, it would be more difficult to show when communication is feasible as well as precisely how it can be achieved.
>     - A complex game would likely have dynamics that are much harder to control (e.g. communication through a visual action space instead of utterances). This would make our arguments and conclusions weaker
> - We believe our results are still interesting despite the simplicity of our game
>     - Previous research has looked at more complex games (e.g. Jaques et al, 2019) but has not quantified the level of cooperation/competition.
>     - We needed to create a game that not only had a quantifiable level of cooperation/competition but also allowed for it to easily tunable, spanning the range of fully cooperative to fully competitive
>     - We believe we have shown the feasibility of achieving communication under competition, setting an example for future research to use better baselines and better quantify the level of competition/cooperation
>
> Loss Function Notation
>
> - We use standard ML notation with $L$ referring to a generic loss function and $L_1$ specifically referring to the absolute distance loss function. Please let us know if this was not clear.
>
> L2 as a hyperparameter metric
>
> - The game with bias = 180 is indeed constant sum (see our proof in the appendix).
>     - Is there a difference between “general constant sum” and “constant sum”?
> - We are indeed implicitly biasing towards fairness by using the $L_2$ metric
>     - We believe this is reasonable to recover the difference between “communication” and “manipulation” because fair communication cannot be “manipulation”. We are open to other ways of achieving that if you have suggestions.
>     - We also show both agents' $L_1$ losses for all hyperparameter search runs in Figure 4 to allow readers to understand the distribution of results on top of just the best hyperparameters we picked
>
> --- continued below ---

---

> > ### Comment · AnonReviewer2 · 2019-11-11
> > **Comments part 3**
> >
> > @  2. “Iteration in the parameter space”.
> > This is fundamentally different from an iterated game. Interesting reciprocity requires agents to be able to respond to the actions of other agents within the episode.
> >
> > @- The loss is not differentiable wrt to the sender:
> > Well, it clearly is - otherwise you would not be able to optimize it using policy gradient (which does nothing but estimate the gradient). My point is that the game is small and simple enough that you don't have to use policy gradient at all. You can simply marginalize across the possible messages and calculate the exact expected return as a function of the weights of the 2 agents. This would not only allow for easier reproducibility but also alleviate that concerns around hyper-parameters etc.
> >
> >
> > @"This would be interesting if the game was complex":
> > This goes back to my point about 'self-interested' being meaningless outside a specific task.
> > We can trivially construct a large number of simple games in which all good strategies / equilibria clearly involve communication (one of which you constructed here).
> > So by construction, if learning works properly, the agents will learn to communicate in these settings.
> > Therefore, the only challenge here is learnability. Indeed - previous claims about 'emergent communication' amongst self-interested agents have been made in complex settings.
> > We know that SOTA MARL works (ie. can find equilibria) in small games, so a toy task teaches us very little.
> >
> > @"- We use standard ML notation with  referring to a generic loss function and  specifically referring to the absolute distance loss function. Please let us know if this was not clear."
> >
> > I don't think you do, at least not consistently:  "$L^i = L(a, T^i)$. By using an $L_1$ loss between the angle of the target and
> > action $L^i_1(T^i, a) = min(|T^i−a|, 360 −|T^i−a|)$"
> >
> > @. We are open to other ways of achieving that if you have suggestions.
> >
> > Yes - get rid of the extensive hyperparamter tuning. The game is simple enough you can do exact gradients and you should not need tuning. This game does not require Deep-RL at all and can entirely be done in a tabular setting.
> >
> > @Confusing Graphs and Error Bars
> > None of the graphs mention what the shading is. Is this the standard error of the mean? If so, why is it so large in Figure 3 c)? Figure 2 d) also points to instability in your training process.
> > Again - the game is simple enough that there should be no question about the training being reliable.

---

> > > ### Author Response · Authors · 2019-11-15
> > > **Marginalizing and Simplicity Experiments**
> > >
> > > Interesting Reciprocity
> > > We agree it is different and reciprocity is indeed interesting. We think that learning to agreeing to a protocol under competition is also interesting
> > >
> > > Differentiability
> > > I think we may have slightly different meanings for “differentiable”. In RL terminology, gradient estimators are used specifically when something is not differentiable. If it were differentiable, we wouldn’t need RL and would just use the exact gradient. And
> > >
> > > Does Marginalizing Across Messages Help?
> > > Marginalizing across the messages is possible and it is just about allowing the sender to backpropogate through the receiver. We coded the marginalizing sender (https://controlc.com/4b0f1f50 ) but after running found no significant difference to the regular reinforce setup (https://pasteboard.co/IGPUlSC.png ) which implies that REINFORCE is a good enough gradient estimator and the instabilities are not from gradient estimation. Simply calculating the return as a function of the two agents isn't sufficient to get to equilibria, you must, at minimum, also take into account the learning dynamics (e.g. LOLA)
> > >
> > >
> > > Learnability
> > > I think we’re very much on the same page about learnability, that is what we’re most interested in. The question is whether “learning works properly” as you state. Previous works have not managed to make it “work properly” and we wanted to demonstrate that it could!
> > >
> > > Cao et al argue that you need prosocial agents, we show that you don't.  Jaques et al argue that you need their SOTA learning rule for selfish communication, but we show that you regular RL is sufficient given the right setup. We think a toy task is just the scenario to show these details and carefully investigate why previous approaches may have been unsuccessful. We believe it is necessary so future work doesn’t automatically assume you need SOTA or fully cooperative agents to emerge communication.
> > >
> > > Notation
> > > We think that notation is consistent. We explain generally that agents get some loss between their target and the action ($L$) and then we specify that we choose this loss to be an $L_1$ loss on the circumference of the circle.
> > >
> > > Hyperparameters Tuning
> > > We think that hyperparameter searches are generally done in all of machine learning and especially deep learning which is very sensitive to it. Our situation even more strongly demands hyperparameter search as we are not just looking to find the “best” model but make arguments about whether something is feasible or not. If we do not do an exhaustive search, we cannot argue that something is infeasible (e.g. communication in high competition)
> > >
> > > Finally, the whole question is how to resolve the issue of communication vs manipulation (we use $L_2$). Your suggestion to “use exact gradients” does not address that because our situation is not that simple (see below)
> > >
> > > Is Our Setup Too Simple?
> > > Looking at similar situations (e.g. https://github.com/facebookresearch/EGG/tree/master/egg/zoo/language_bottleneck/guess_number ) and the architectures present there, our game is more complex but our architectures are comparable.
> > >
> > > Graphs
> > > The shading is indeed the standard error of the mean. This is in the description for Figure 2 but we will add it to the other descriptions as well to make it more clear.
> > >
> > > Error Bars
> > > Given that even fully cooperative emergent communication does not converge occasionally, and our situation is complicated by divergent interests, we don’t think our graphs are too high variance. Each point is a different experiment where 5 random seeds run for a particular bias, so the graphs definitely look higher variance than usual RL graphs charting reward over time. We are honest with our random seeds and make no attempt at tuning them. At the very least, our results are significant despite the variance.

---

> ### Author Response · Authors · 2019-11-11
> **Reviewer 2 Response Part 4**
>
> Broken Reference
>
> - We thank the reviewer for pointing this out and will fix this to read Figure 2
>
> Experiments for Bias = 180
>
> - Since we are plotting the sum of rewards, the curve would be trivially at 180.
> - We could plot each agent’s individual reward. A couple test runs of our basic setup were found to have non-communication (90/90) error split
> - If you have specific experiments and plots you would like us to make, we would be happy to investigate
>
> References:
>
> Kharitonov et al. “EGG: a toolkit for research on Emergence of lanGuage in Games” Arxiv 2019. https://github.com/facebookresearch/EGG/
>
> Kingma, Diederik P. and Max Welling. “Auto-Encoding Variational Bayes.” ICLR (2014)
>
> Lanctot, Marc et al. “A Unified Game-Theoretic Approach to Multiagent Reinforcement Learning.” NIPS (2017).
>
> Resnik, David B. “How-Possibly Explanations in Biology”. Acta Biotheoretica (1991) ,39(2):141–149.
>
> Singh, Satinder P. et al. “Nash Convergence of Gradient Dynamics in General-Sum Games.” UAI (2000).
>
> Shoham, Yoav et al. “Multi-Agent Reinforcement Learning: A Critical Survey.” (2003).

---

> > ### Comment · AnonReviewer2 · 2019-11-11
> > **response part 4**
> >
> > @- We could plot each agent’s individual reward. A couple test runs of our basic setup were found to have non-communication (90/90) error split
> >
> > Yes - please include this for completeness.
> >
> > @- If you have specific experiments and plots you would like us to make, we would be happy to investigate
> >
> > Sure, there are 3 interesting cases that are missing from the paper:
> > 1) Exact gradient version of the game in a tabular setting. This will rule out any learning issues.
> > 2) Iterated game with cheap talk. Can agents learn to use the cheap talk channel to reciprocate / communicate? If so, which components of modern MARL are required for this? Note that this should be done in a setting where the cheap-talk channel allows for interesting reciprocity. Ie. I can tell you the truth now or lie to you and then you can reciprocate during the next time step.
> > 3) Go back to the negotiation setting (https://arxiv.org/abs/1706.05125) and show that with SOTA MARL self-interested agents can indeed learn to use the cheap-talk channel. This would be a result.  Equally, properly understanding why agents do not learn to use the cheap-talk channel would be interesting.
> >
> > After carefully reading through the author response I do not believe that the paper in the current form is ready for publication, but if those experiments were carried out I would be happy to reconsider.

---

> > > ### Author Response · Authors · 2019-11-13
> > > **2/3 Interesting Experiments Response**
> > >
> > > Tabular Exact Gradient
> > > - We can create exact gradients by having the message be continuous and allow the sender to backpropagate directly through the receiver.
> > >     - This is functionally similar to your suggestion about “marginalizing across messages” but cleaner to implement
> > >     - This is also just the continuous game with a more powerful sender that essentially has one-sided access to the receiver’s parameters
> > >     - We had a couple experiments but did not pursue them deeply because we preferred a more symmetric setup. We will take a closer look
> > > - Because our state space is continuous, we cannot do tabular RL but instead choose to use policy gradient. Would you want to see a discrete state space instead of a continuous one?
> > >     - This should not improve learning stability and may actually harm it because discrete spaces are not ordered by default
> > >     - Our agents would need to learn the ordering of the space (e.g. input 1 < input 2) as well as how to use it
> > >     - Preliminary experiments we made with discrete action spaces when designing the game were not promising
> > >
> > > Iteration
> > > - We did actually implement an iterative game in exploratory experiments.
> > > - It is more difficult to design and train effectively because we had to add a notion of statefulness
> > >     - One option is to use an RNN to maintain state but this complicates learning dynamics
> > >     - Another option is to condition on a single previous state (as in LOLA) but the two-step nature of our game making it difficult to specify an unbiased initial state. Agents need to distinguish the first state as having no history but it is not straightforward to have a null previous state for our continuous state space
> > > - Training was less stable and learning dynamics were more complicated.
> > >     - Ultimately, it was much more computationally intensive which made our extensive hyperparameter searches much less feasible to run.
> > >     - Sadly, we could not achieve baselines we felt to be reasonable
> > > - The idea of alternating lies and truth over rounds is interesting but “lying” is not as straightforward as it seems
> > >     - Because emergent communication is not separately learning a meaning and use but learning the two simultaneously
> > >     - It can be hard to learn what is “lying” because it relies on there being an existing meaning and subverting that meaning
> > >     - Learning agents would not be able to distinguish between lying and misinterpreting a signal and would just adjust their distribution of meanings ("without somewhat agreeing to meanings, agents cannot use those meanings to compete (Searcy & Nowicki, 2005; Skyrms & Barrett, 2018).")
> > > - We chose to focus on the initial problem of learning to communicate honestly but noisily

---

> > > > ### Comment · AnonReviewer2 · 2019-11-14
> > > > **Response**
> > > >
> > > > @Exact gradients:
> > > > I think it would be easy to discretize the state space without changing the fundamental nature of the game (in particular given that there already is a discrete communication channel).
> > > >
> > > > @Iteration:
> > > > Yes - that's partially what I am trying to say. It is harder but (I believe) also a lot more interesting from a research point of view.  The initial state is easy to deal with in the discrete version of the game, which I would recommend either way. In the discrete version the initial state just becomes a different one-hot.
> > > > I agree that truth / lying can only ever be defined given the 'convention' that has been established across the channel.

---

> > > ### Author Response · Authors · 2019-11-13
> > > **Negotiation Game Experiment**
> > >
> > > It's interesting you mention the negotiation game because figuring out why communication didn't emerge there was the starting point of our research
> > >
> > > - We could not reproduce all the curves for selfish agents seen in Cao et al (2018)
> > >     - We contacted the authors and together still did not manage to reproduce their curve for “Proposal” “Linguistic” and “Both”
> > >     - We could not figure out why their agents performed more fairly in the presence of a linguistic channel (“Linguistic”) despite one agent dominating in the non-communication case and therefore having no incentive to communicate
> > >     - We explained our results and verified all differences we could think of but the authors could not suggest a reason for the difference
> > >     - Deepmind did not publicly release their code
> > >
> > > - Instead, we found the first player to be regularly dominating the second player because of learning dynamics
> > >     - The first player could not make an accept action on their first move which made it more likely that the second player was the first to learn the accept action
> > >     - The second player would see two possible outcomes if the accept action could succeed: 1. Gain whatever reward was associated with accepting the deal 2. Continue negotiating and likely go past the final round and end up with no reward
> > >     - Once the second player was constantly accepting the deal given to it, the first player would learn to give it a deal with as little reward as possible
> > >     - We found this to be a stackelberg game where the second player has little recourse if they do not understand the nature of the game. The first player would give the second one a deal and the second player would just accept it
> > >
> > > - The main obstacle to communication was the domination of one agent as well as the highly competitive nature of the game
> > >     - Since the first agent dominated the other under non-communication, it was never in their interest to communicate
> > >     - We allowed agents to mask their communication and found that the dominating agent always masked their communication
> > >     - We achieved some communication when we did partial reward sharing (Peysakhovich and Lerer, 2017) but found that we still needed to tune hyperparameters to disadvantage the first agent
> > >     - If the reward sharing was high enough, it became the first agent’s best interests to communicate and communication emerged.
> > >
> > > - We eventually decided that the negotiation game was not a good test bed for these experiments because the learning dynamics allowed one agent to dominate
> > >     - A simple idea, inspired by Lowe et al (2019), is that communication should only emerge if the possible reward under communication is greater than under non-communication for both agents
> > >     - Since the negotiation game has one agent dominating and the game is often zero-sum, we do not think any algorithm can lead to communication
> > >     - We decided to come up with a different game to specifically look at the role of cooperation/communication unfettered by common issues in emergent communication games: communication through the action space, unquantified game dynamics, unquantified optimal possible play, and badly tuned baselines.
> > >
> > > Sorry for going over the 5000 character limit with these two posts but we wanted to give more detail on the results of the experiment you were asking about.

---

> > > > ### Comment · AnonReviewer2 · 2019-11-14
> > > > **Reviewer response**
> > > >
> > > > @It's interesting you mention the negotiation game because figuring out why communication didn't emerge there was the starting point of our research:
> > > >
> > > > I am glad to hear that this was the starting point. I believe a thorough investigation into this topic and improvement of the setting to address the issues you find would indeed make for a very interesting contribution.
> > > >
> > > > @" We found this to be a stackelberg game where the second player has little recourse if they do not understand the nature of the game."
> > > >
> > > > I am not sure what this sentence is supposed to mean. What does it mean for an RL agent to 'understand' the nature? I am going to come back to the point that at the end of the day it's about the possible equilibria of the game. In particular I still don't understand whether this is an issue with the game design (all good strategies are trivial) or the learning methods.
> > > >
> > > > @"highly competitive":
> > > > This is not inherent in the game setup. You can easily imagine that the payouts agents obtain for different objectives are different enough such that cooperation is more encouraged.
> > > >
> > > > @ "the game is often zero-sum":
> > > > The game is most likely rarely zero-sum (both agents would have to have the same rewards per item).
> > > >
> > > > @5000 character limit:
> > > > It is good to be thorough, but there also is value in distilling thoughts and arguments into concise form ("If I had more time, I would have written a shorter letter").

---

> > > > > ### Author Response · Authors · 2019-11-15
> > > > > **Negotiation Game Followup**
> > > > >
> > > > > You’re right, to emerge communication in the negotiation game we should carefully construct the sampling of agent preferences (weights) and sampling of items to be more cooperative. This would indeed be a nice addition to the paper and we would be happy to add it to the camera-ready if we manage to overcome the learning dynamic issues of the game.
> > > > >
> > > > > By “understand the nature of the game” we mean that RL agents must discover the equilibria but given the game setup and learning dynamics, one agent learns to dominate early on. The dominating agent can then prevent the other from discovering better options and the learning dynamics of RL can make it infeasible for a badly losing agent to recover in this game.
> > > > >
> > > > > The game is indeed not zero-sum, we meant to write "fully competitive" (unless one agent has weight 0 for an item, agents are competing on all items, there is no common reward they can optimize together, they must always compromise)
> > > > >
> > > > > @Concision: Agreed.

---

### Author Response · Authors · 2019-11-11
**Crawford and Sobel Response**

Since both Reviewer 1 and Reviewer 2 have brought up issues related to how our work differs from Crawford and Sobel (1986), we thought we would address them together and try to clarify the differences.

Crawford and Sobel do a static analysis at equilibria and give existence proofs and guarantees. In contrast, we do a dynamic analysis and focus on showing empirical feasibility of competitive selfish communication in the modern ML paradigm of emergent communication.


Static Analysis vs Dynamic Analysis

- Crawford and Sobel study possible equilibria and the fixed communication protocols at those equilibria.
    - They prove the existence of equilibria and their properties but do not show how to achieve those equilibria nor whether certain learning rules could lead to and maintain those equilibria
- We specifically study the standard setup used in emergent communication: learning by gradient descent on the receiver and REINFORCE (or variants) on the sender. We show that even in our basic scenario, it is feasible to achieve communication when the game is more cooperative than competitive.
    - Our agents are also not seeking equilibria when they achieve communication, they are simply seeking selfish reward without taking the opponent into mind

- Crawford and Sobel show that communication is not possible after a degree of divergence in interests
- We empirically demonstrate that emergent communication is feasible with regular agents in the modern paradigm, overturning a previous misconception.
    - We quantify the exact circumstances (level of competition) that would cause this misconception (when the game is more competitive than cooperative)
    - We make a strong case for all future papers in competitive emergent communication to precisely quantify the level of competitiveness in the game (something that is not currently done e.g. Leibo et al (2017)). This would give a better perspective on the efficacy of achieved and achievable communication.

Knowledge of the Game and Opponent

- Crawford and Sobel suppose that agents are “perfectly rational” and are given perfect knowledge (see Shoham et al 2003)
    - Both players are fully aware of the nature of the game, and have knowledge of their and the other player’s reward for all situations, and always take the rational best response.
    - Messages are modelled as states of the world with added noise
- We start from the assumption of “bounded rationality” and only add modelling of the opponent and game as necessary
    - We suppose nothing about an agent’s knowledge of the other player, the rules of the game, or how they should act.
    - Our agents use RL to discover all knowledge through trial and error, with the goal of optimizing their own reward. Our agents do not see the other player’s reward and are solely optimizing their own.
    - We only add opponent modelling as necessary for LOLA. Agents use a model of their opponent and built into LOLA is the assumption that opponents learn with gradient descent and have similar loss objectives.
    - Messages are simply mappings of the world to symbols and do not necessarily need to be ordered or completely cover all states of the world


In short, our work seeks to offer a fundamental contribution to the field of emergent communication and machine learning. We are leaning on the work of Crawford and Sobel as a guide for possible equilibria but fundamentally we wish to show feasibility with learning dynamics not possibility and theoretical guarantees. Our hope with this work is to correct a misconception about selfish emergent communication prevalent in the field, to bring the theoretical contributions of Crawford and Sobel into the fold of emergent communication literature, and to give guides about how to correctly measure and possibly improve emergent communication under competition.


References:

Leibo, Joel Z. et al. “Multi-agent Reinforcement Learning in Sequential Social Dilemmas.” AAMAS (2017).

Shoham, Yoav et al. “Multi-Agent Reinforcement Learning:a critical survey.” (2003).

---

### Author Response · Authors · 2019-11-15
**Paper Update and Thanks**

We’ve updated the paper based on the discussions here:

- Fixed Figure reference
- Changed papers cited for claim that previous work didn’t find selfish communication
- Rewritten our section on Crawford and Sobel to reflect this discussion
- Added results for bias = 180 in the appendix for completeness

We want to thank all the reviewers for their comments and responses, especially Reviewer 2 with whom we’ve had an in-depth and productive discussion. Though there are still disagreements, we are deeply thankful for their responsiveness and willingness to interact with our work.

Thank you

---

### Decision · Program_Chairs · 2019-12-19

**Decision:**

Reject

**Comment:**

There has been a long discussion on the paper, especially between the authors and the 2nd reviewer. While the authors' comments and paper modifications have improved the paper, the overall opinion on this paper is that it is below par in its current form. The main issue is that the significance of the results is insufficiently clear.  While the sender-receiver game introduced is interesting, a more thorough investigation would improve the paper a lot (for example, by looking if theoretical statements can be made).